# Area 2 of primary somatosensory cortex encodes kinematics of the whole arm

**Raeed H Chowdhury[1,2]\*, Joshua I Glaser[3,4,5], Lee E Miller[1,6,7,8]\***

[1]Department of Biomedical Engineering, Northwestern University, Evanston, United States; [2]Systems Neuroscience Institute, University of Pittsburgh, Pittsburgh, United States; [3]Interdepartmental Neuroscience Program, Northwestern University, Chicago, United States; [4]Department of Statistics, Columbia University, New York, United States; [5]Zuckerman Mind Brain Behavior Institute, Columbia University, New York, United States; [6]Department of Physiology, Northwestern University, Chicago, United States; [7]Department of Physical Medicine and Rehabilitation, Northwestern University, Chicago, United States; [8]Shirley Ryan AbilityLab, Chicago, United States

**Abstract** Proprioception, the sense of body position, movement, and associated forces, remains poorly understood, despite its critical role in movement. Most studies of area 2, a proprioceptive area of somatosensory cortex, have simply compared neurons' activities to the movement of the hand through space. Using motion tracking, we sought to elaborate this relationship by characterizing how area 2 activity relates to whole arm movements. We found that a whole-arm model, unlike classic models, successfully predicted how features of neural activity changed as monkeys reached to targets in two workspaces. However, when we then evaluated this whole-arm model across active and passive movements, we found that many neurons did not consistently represent the whole arm over both conditions. These results suggest that 1) neural activity in area 2 includes representation of the whole arm during reaching and 2) many of these neurons represented limb state differently during active and passive movements.

**\*For correspondence:**
raeed.chowdhury@gmail.com (RHC);
lm@northwestern.edu (LEM)

**Competing interests:** The authors declare that no competing interests exist.

## Introduction

Moving in an uncontrolled environment is a remarkably complex feat. In addition to the necessary computations on the efferent side to generate movement, an important aspect of sensorimotor control is processing the afferent information we receive from our limbs, essential both for movement planning and for the feedback it provides during movement. Of the relevant sensory modalities, proprioception, or the sense of body position, movement and associated forces, is arguably the most critical for making coordinated movements (*Ghez and Sainburg, 1995*; *Gordon et al., 1995*; *Sainburg et al., 1995*; *Sainburg et al., 1993*; *Sanes et al., 1984*). However, despite its importance, few studies have explicitly addressed how proprioception is represented in the brain during natural movement; touch, vision, and the motor areas of the brain have received far more attention.

One brain area likely important for mediating reach-related proprioception is the proximal arm representation within area 2 of primary somatosensory cortex (S1) (*Jennings et al., 1983*; *Kaas et al., 1979*; *London and Miller, 2013*). Though this area receives a combination of muscle and cutaneous inputs (*Hyvärinen and Poranen, 1978*; *Padberg et al., 2019*; *Pons et al., 1985*), the few studies examining it during reaching have found that a model involving simply the translation of the hand approximates neural activity quite well (*London and Miller, 2013*; *London et al., 2011*; *Prud'homme and Kalaska, 1994*; *Weber et al., 2011*). Interestingly, this finding fits with psychophysical data showing that humans are better at estimating the location of the hand than joint angles (*Fuentes and Bastian, 2010*), as well as our conscious experience of reaching to objects, which typically focuses on the hand. However, recent computational studies have shown that while neural

activity may appear to be tuned to the state of a limb's endpoint, features of this tuning might be a direct consequence of the biomechanics of the limb (*Chowdhury et al., 2017*; *Lillicrap and Scott, 2013*). Consistent with those results, we have recently observed, using artificial neural networks, that muscle lengths were better predictors of area 2 activity than were hand kinematics (*Lucas et al., 2019*).

As in the classic reaching studies of M1 (*Caminiti et al., 1991*; *Georgopoulos et al., 1982*; *Georgopoulos et al., 1986*), the appeal of the hand-based model of area 2 neural activity is its reasonable accuracy despite its simplicity. However, the recent emphasis on studying less constrained, more natural movements (*Mazurek et al., 2018*) is pushing the limits of such simple models (*Berger and Gail, 2018*; *Hasson et al., 2012*; *Sharon and Nisky, 2017*). As in the motor system, it is increasingly important to characterize proprioceptive regions' responses to reaching more fully. Here, we used two experiments that altered the relationship between hand and whole-arm kinematics. In the first experiment, we found that neurons in area 2 have a consistent relationship with whole-arm kinematics during active reaching within two disjoint workspaces. Whole-arm kinematics predicted neural activity significantly better than did the hand-only model, and were able to effectively explain neural activity changes across workspaces. In the second experiment, we compared area 2 responses to active reaching and passive perturbations of the hand. While some neurons were predicted well with only kinematic inputs, others were not, adding to the evidence that area 2 may receive efferent information from motor areas of the brain (*London and Miller, 2013*; *Nelson, 1987*).

## Results

For the experiments detailed in this paper, we recorded neural signals from three Rhesus macaques (Monkeys C, H, and L) using Utah multi-electrode arrays (Blackrock Microsystems) implanted in the arm representation of Brodmann's area 2 of S1 (*Figure 1*). After implantation, we mapped sensory receptive fields of each neuron, to examine how the multi-unit activity on each electrode responded to sensory stimulation, noting the modality (deep or cutaneous) and location of each field. We classified an electrode as 'cutaneous' if we could find a receptive field on the arm or torso in which brushing the skin caused an increase in activity. 'Deep' electrodes were those that responded to joint movement or muscle palpation and did not appear to have a cutaneous receptive field. With these criteria, it is likely that some of the electrodes we marked cutaneous actually responded to both deep and cutaneous stimuli. However, as we were most interested in the distribution of receptive field types over the array, we did not test for such mixed modality neurons.

*Figure 1* shows the resulting sensory maps from the mapping session for each monkey in which we were able to test the most electrodes. We found both deep and cutaneous receptive fields across each array, largely matching the description of area 2 from previous studies (*Hyvärinen and Poranen, 1978*; *Pons et al., 1985*; *Seelke et al., 2012*). Of the two bordering regions, area 1 tends to have a higher fraction of cutaneous responses, and area 5 tends to have a higher fraction of deep responses (*Seelke et al., 2012*), suggesting that our arrays were implanted largely in area 2. For Monkeys C and H, we found a rough proximal to distal arm gradient, running from anterior to posterior across the array (*Figure 1*, black arrows), consistent with the somatotopy found by *Pons et al. (1985)*. There were too few well-mapped neurons from Monkey L to determine a meaningful gradient.

We trained each of these monkeys to grasp a two-link planar manipulandum and make reaching movements to targets presented on a screen in front of them (*Figure 2*). During these sessions, we collected interface force from a six degree of freedom load cell attached to the manipulandum handle. We also tracked the locations of markers on the monkey's arm using a custom motion tracking system based on a Microsoft Kinect. Our experiments included two components: one comparing reaching movements in two different workspaces and one comparing active and passive movements.

### Somatosensory area 2 represents the movement of the whole arm during reaching

Previous literature has characterized area 2 primarily in terms of the hand's trajectory through space (*London and Miller, 2013*; *Prud'homme and Kalaska, 1994*; *Weber et al., 2011*), likely in part due

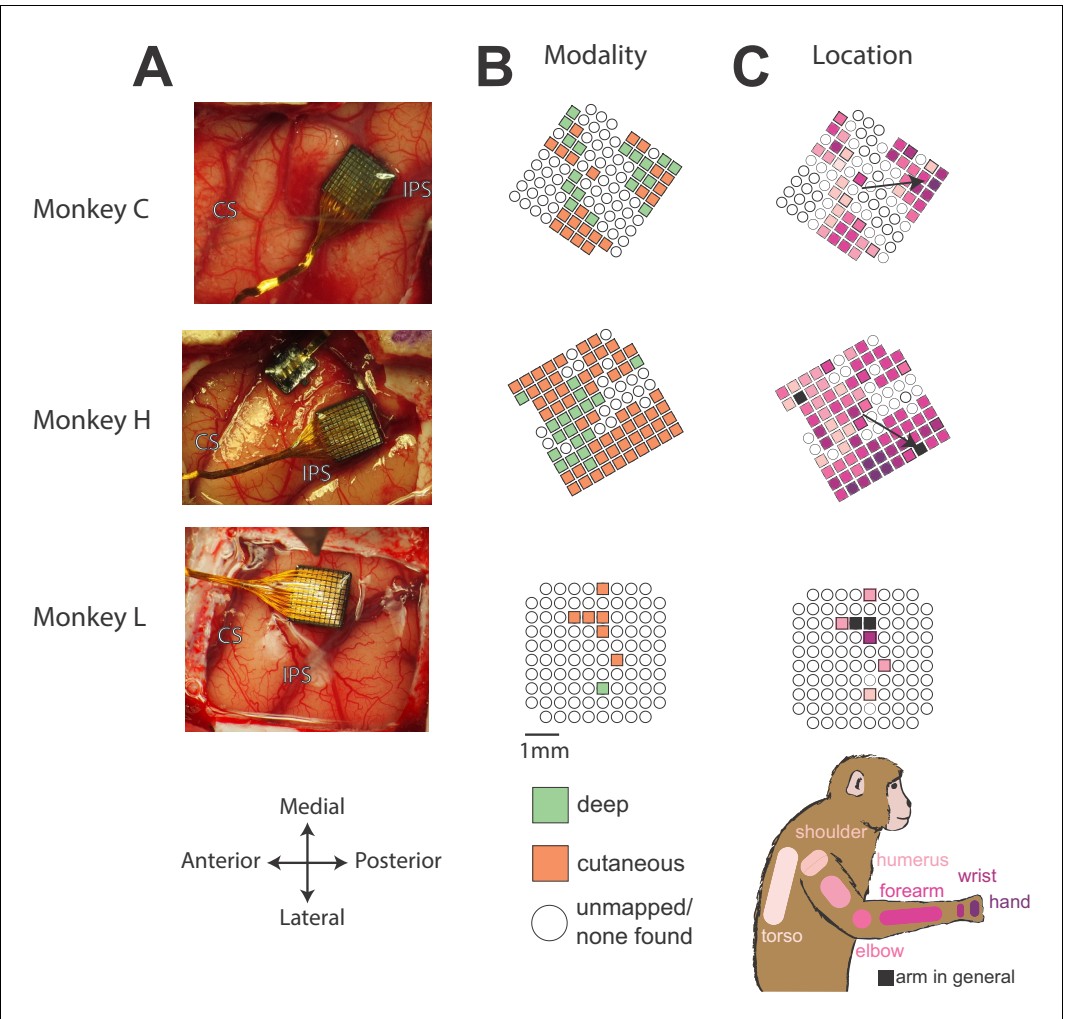

**Figure 1.** Array locations and receptive field maps from one mapping session for each monkey. (**A**) Locations of Utah arrays implanted in area 2 of Monkeys C, H, and L. IPS, intraparietal sulcus; CS central sulcus. (**B**) Map of the receptive field modality (deep or cutaneous) for each electrode. (**C**) Map of receptive field location (see legend on bottom right). Open circles indicate both untested electrodes and tested electrodes with no receptive field found. Black arrows on maps in C show significant gradient across array of proximal to distal receptive fields (see Materials and methods).

to the difficulty of tracking the motion of the full arm, and the then recent finding that motor cortex could be well explained simply by the direction of hand movement (*Caminiti et al., 1991*; *Georgopoulos et al., 1982*). Given advances in motion tracking capability and subsequent observations of the dependence of M1 on arm posture (*Morrow et al., 2007*; *Scott and Kalaska, 1995*), we set out to characterize more fully, how neural activity in area 2 corresponds to reaching movements.

In particular, we aimed to characterize how much could be gained by using models incorporating the movement of the whole arm, as opposed to just the hand. A challenge in comparing these models is that for the typical, center-out reaching task in a small workspace, the behavioral signals used in our models are highly correlated. Because a high correlation means that a linear transform can accurately convert one set of signals into another, all models would make very similar predictions of neural activity.

To deal with this problem, we trained the monkeys to reach to randomly-generated targets presented in two different workspaces (*Figure 3*). This had two important effects. First, the random locations of the targets lessened the stereotopy of the movements, allowing for the collection of more varied movement data than from a center-out paradigm. Second, the average postures in the

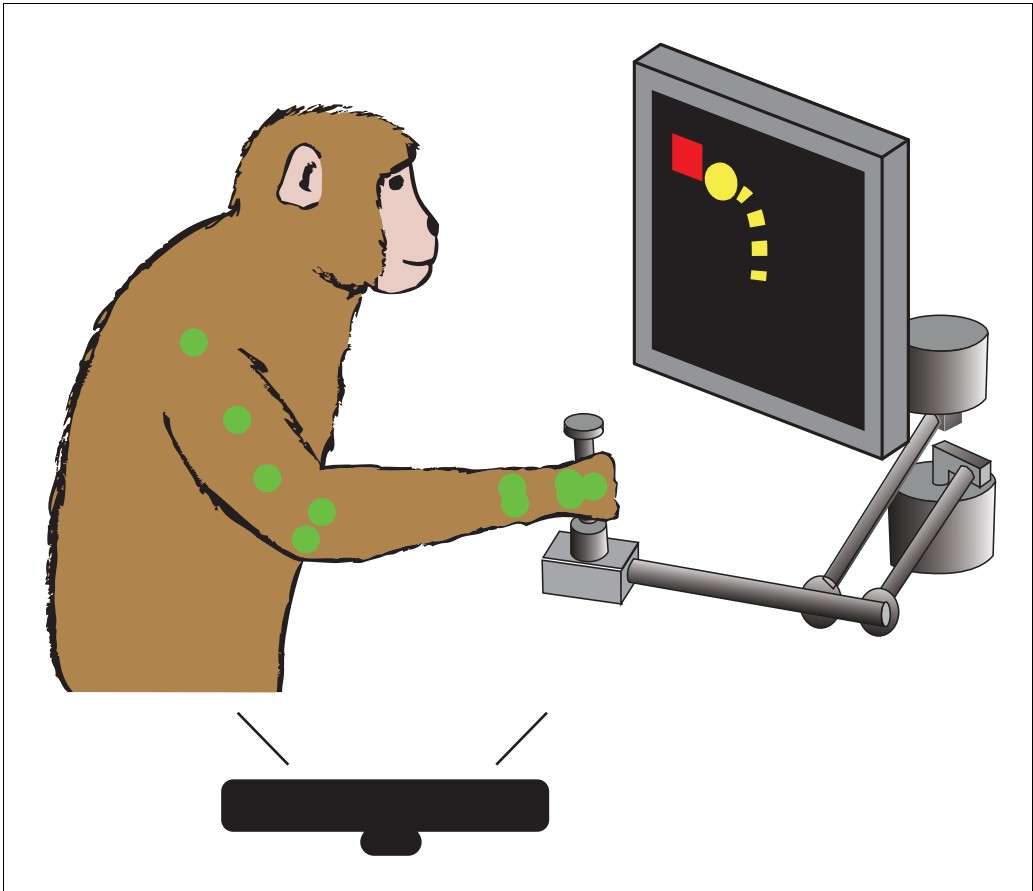

**Figure 2.** Behavioral task. Monkey controls a cursor on screen (yellow) with a two link manipulandum to reach to visually presented targets (red). We track the locations of different colored markers (see Materials and methods) on the monkey's arm (here shown green) during the task, using a Microsoft Kinect.

two workspaces were quite different, such that while the signals of different models were still correlated within a given workspace, this correlation (and the mapping between sets of behavioral signals) changed significantly between workspaces. This forced the models to make different predictions of neural activity across the two workspaces. By comparing modeled and observed changes in neural activity, we could more reliably discriminate between models.

This idea is exemplified in *Figure 3D*. When tested in the two workspaces, this example neuron changed both its tuning curve and the direction in which it fired maximally (its preferred direction, or PD), as did many neurons we recorded. The corresponding predictions of the hand-only and whole-arm models differed, which allowed us to compare the accuracy of the two models. We recorded three of these two-workspace sessions with each of Monkeys C and H and two sessions with Monkey L.

## Model overview

We tested several kinematic models of area 2 activity that could be divided into hand-only and whole-arm models (see Materials and methods for a full description of all the models). We've chosen to represent the two sets with two of the models, which we termed, for simplicity, the 'hand-only' and 'whole-arm' models. The hand-only model stems from classic, endpoint models of limb movement-related neural activity (*Bosco et al., 2000*; *Georgopoulos et al., 1982*; *Prud'homme and Kalaska, 1994*). It assumes neurons relate only to the Cartesian coordinates of hand position and velocity. The whole-arm model builds on the hand-only model by adding the Cartesian kinematics (position and velocity) of the elbow, in order to account more fully for movement of the whole arm. Surprisingly, the performance of this representation of the whole arm was similar to, or even better

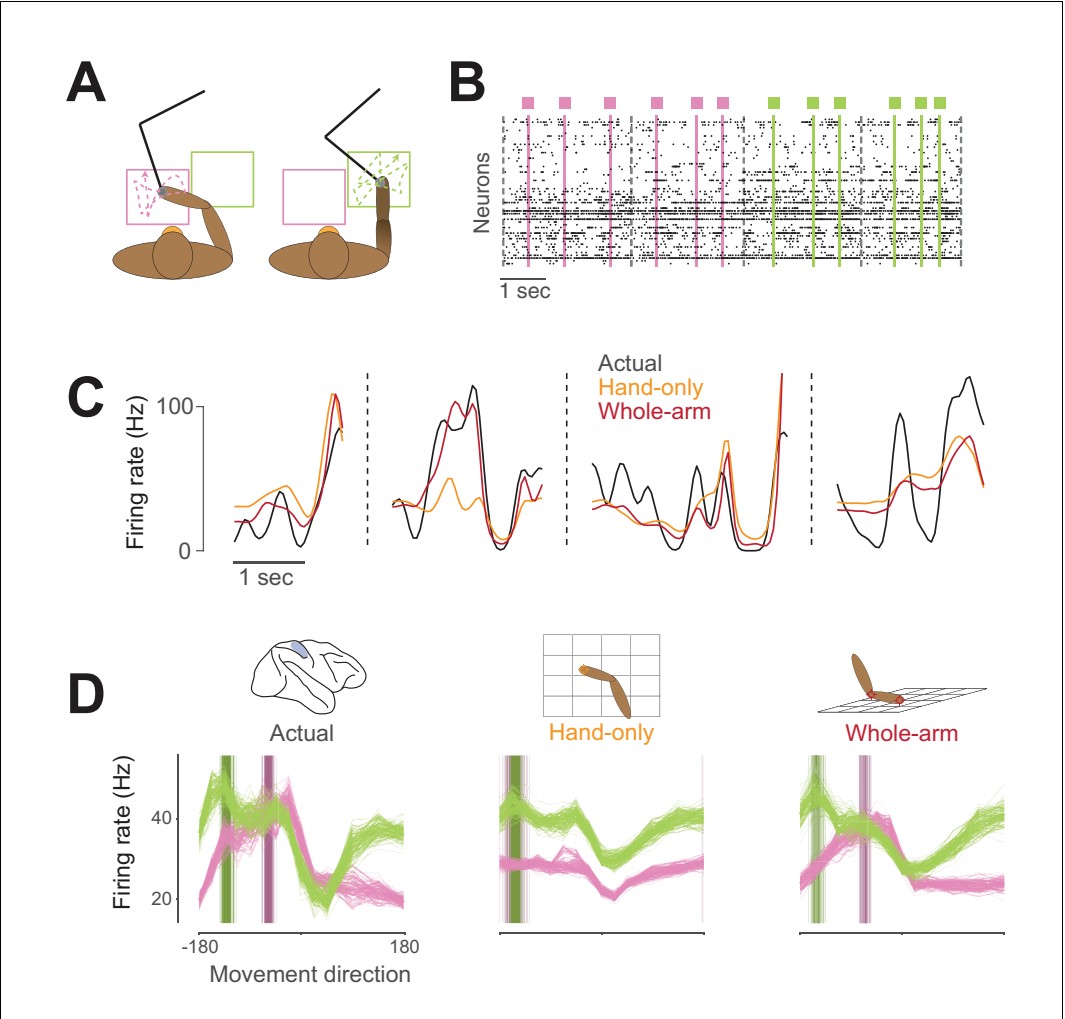

**Figure 3.** Example neural activity for two-workspace task. (**A**) Two-workspace behavior. On each trial, monkey reaches with manipulandum (black) to randomly placed targets in one of two workspaces: one close to the body and contralateral to the reaching hand (pink) and the other distant and ipsilateral (green). Trials in the two workspaces were interleaved randomly. (**B**) Example neural raster plot from one session for two randomly drawn trials in each workspace. Dots in each row represent activity for one of the simultaneously recorded neurons. Black dashed lines indicate starts and ends of trials, and colored lines and boxes indicate times of target presentation, with color indicating the workspace for the trial. (**C**) Firing rate plot for an example neuron during four randomly drawn trials from the distal (green) workspace. Black trace represents neural firing rate, smoothed with a 50 ms Gaussian kernel. Colored traces represent unsmoothed firing rates predicted by hand-only (orange), and whole-arm (red) models. (**D**) Actual and predicted tuning curves and preferred directions (PDs) computed in the two workspaces for an example neuron. Each trace represents the tuning curve or PD calculated for one cross-validation fold (see Materials and methods). Leftmost plot shows actual tuning curves and PDs, while other plots show curves and PDs for activity predicted by each of the models. Each panel shows mean movement-related firing rate plotted against direction of hand movement for both workspaces. Darker vertical bars indicate preferred directions.

than more complicated biomechanical models based on the seven degree-of-freedom joint kinematics or musculotendon lengths (see Appendix 1). We aimed to test how well the hand-only and whole-arm models predicted features of neural activity during reaching, in order to determine the importance of whole-arm kinematics for explaining neural activity.

For us to consider the whole-arm model to be an effective one for area 2, it should satisfy three main criteria. First and most direct, it should explain the variance of neural firing rates across the two workspaces better than the hand-only model, as is the case in the example in *Figure 3C*. Second,

the mapping between neural activity and whole-arm kinematics should not change between the individual workspaces, meaning that the accuracy of a model trained over both workspaces should be similar to that trained in a single workspace. Last, the model should be able to capture features of neural activity that it was not explicitly trained on, for example, the changes in directional tuning shown in *Figure 3D*.

## Whole-arm model explains more variance of area 2 neural activity than hand-only model

To assess how well our models fit area 2 neural activity, we used repeated k-fold cross-validation (see Materials and methods for more details). Goodness-of-fit metrics like $R^2$ or variance-accounted-for (VAF) are ill-suited to the Poisson-like statistics of neural activity; instead, we used the likelihood-based pseudo-$R^2$ (*Cameron and Windmeijer, 1997*; *Cameron and Windmeijer, 1996*; *McFadden, 1977*). Like VAF, pseudo-$R^2$ has a maximum value of 1, but it can be negative for models that fail even to predict the mean firing rate during cross-validation. In general, the values corresponding to a good fit are lower for $pR^2$ than for either $R^2$ or VAF, with a value of 0.2 usually considered a 'good' fit (*McFadden, 1977*). We found that for this measure, the whole-arm model out-performed the hand-only model (*Figure 4*). Of the 288 neurons recorded across the 8 sessions, 238 were significantly better predicted by the whole-arm model than the hand-only model, and for the other 50, there was no significant difference (using $p < 0.05$; see Materials and methods for more details).

## Whole-arm model captures a consistent relationship between area 2 and arm kinematics

A reasonable benchmark of how well the whole-arm model fits the two-workspace data is its ability to match the accuracy of models trained in the individual workspaces. It is possible to imagine a scenario in which a model might achieve a good fit by capturing a global relation across the two workspaces without capturing much information local to either workspace. This situation is akin to fitting a line to data distributed along an exponential curve. In this analogy, we would expect a piecewise linear fit to each half of the data to achieve significantly better goodness-of-fit.

We tested this scenario by training whole-arm models on the individual workspaces, and comparing the resulting $pR^2$ with that calculated from the model fit to data from both workspaces. The symbols lying very close to the unity line in each panel of *Figure 5* indicate that the full model explained just as much neural variance as did the individual models. This suggests that the whole-arm model describes a consistent, generalizable relationship between neural activity and arm kinematics across the two workspaces.

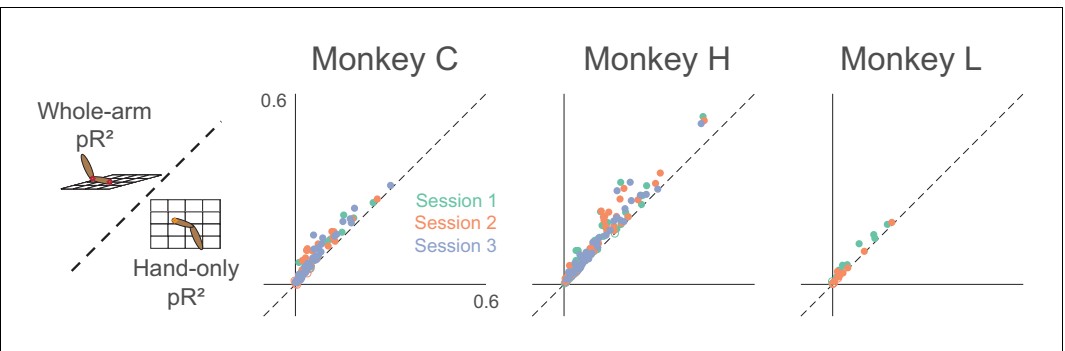

**Figure 4.** Goodness-of-fit comparison analysis. Scatter plots compare the pseudo-$R^2$ ($pR^2$) of the whole-arm model to that of the hand-only model for each monkey. Each point in the scatter plot represents the $pR^2$ values of one neuron, with whole-arm $pR^2$ on the vertical axis and hand-only $pR^2$ on the horizontal. Different colors represent neurons recorded during different sessions. Filled circles represent neurons for which one model's $pR^2$ was significantly higher than that of the other model. In this comparison, all filled circles lie above the dashed unity line, indicating that the whole-arm model performed better than the hand-only model for every neuron in which there was a significant difference.

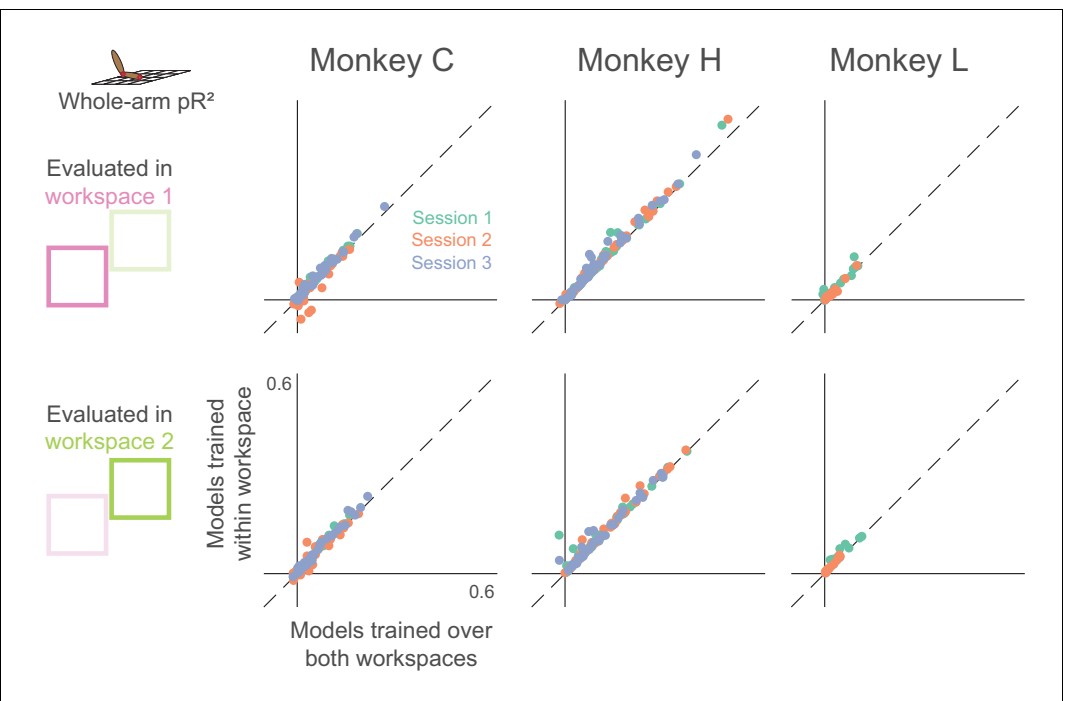

**Figure 5.** Dependence of whole-arm model accuracy on workspace location of training data. Each panel compares a model trained and tested in the same workspace (either near or far) to a model trained on data from both workspaces. Each dot corresponds to a single neuron, where color indicates the recording session. Dashed line is the unity line.

## Whole-arm model captures changes in area 2 directional tuning between workspaces

From previous studies of area 2, we know that at least within a single workspace, neural activity is tuned approximately sinusoidally to the direction of hand movement (*London and Miller, 2013*; *Prud'homme and Kalaska, 1994*; *Weber et al., 2011*). *Figure 3D* shows the directional tuning curves for an example neuron, along with the tuning curves predicted by both models. Because we trained each model on data from both workspaces, they needed to capture a single relationship between movement and neural activity. As shown in the example in *Figure 3D*, the hand-only model predicted essentially the same tuning curve for both workspaces, with the exception of a baseline

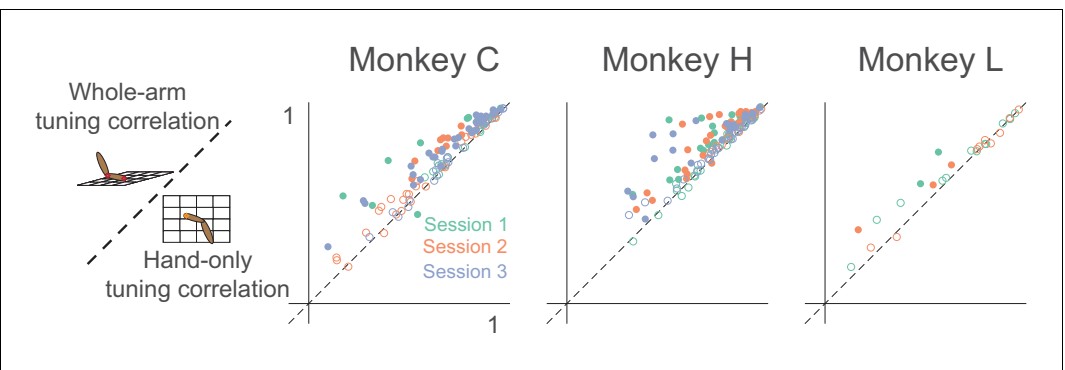

**Figure 6.** Tuning curve shape correlation analysis. Scatter plot compares tuning curve shape correlation between whole-arm and hand-only models. Filled circles indicate neurons significantly above or below the dashed unity line. As for pR$^2$, most filled circles lie above the dashed line of unity, indicating that the whole-arm model was better at predicting tuning curve shape than the hand-only model.

shift due to the position component. In contrast, the whole-arm model predicted altered tuning curves, which matched the actual ones well.

To quantify this model accuracy over all neurons, we calculated the correlation between the model-predicted and actual tuning curves in the two workspaces. With this measure, the whole-arm model once again won most of the pairwise comparisons (*Figure 6*). Only two out of 288 neurons were significantly better predicted by the hand-only model (using p < 0.05), while 138 of 288 neurons were significantly better predicted by the whole-arm model.

Of the 288 recorded neurons, 260 were significantly tuned to movement direction in both workspaces. Thus, in addition to the tuning curve correlation analysis, we also examined the PD in the two workspaces. For many neurons, the PD changed significantly between workspaces, as in the leftmost panel of *Figure 3D*. *Figure 7A* shows the actual PD shifts for all neurons plotted against the PD shifts predicted by each model. The large changes in PD, shown on the horizontal axes of the scatter plots, are a clue that the hand-only model does not fully account for area 2 neural activity; if it had, the PD changes should have been insignificant (in principle, zero), as shown by the generally small hand-only model-predicted changes (first row of *Figure 7A*). Additionally, and perhaps

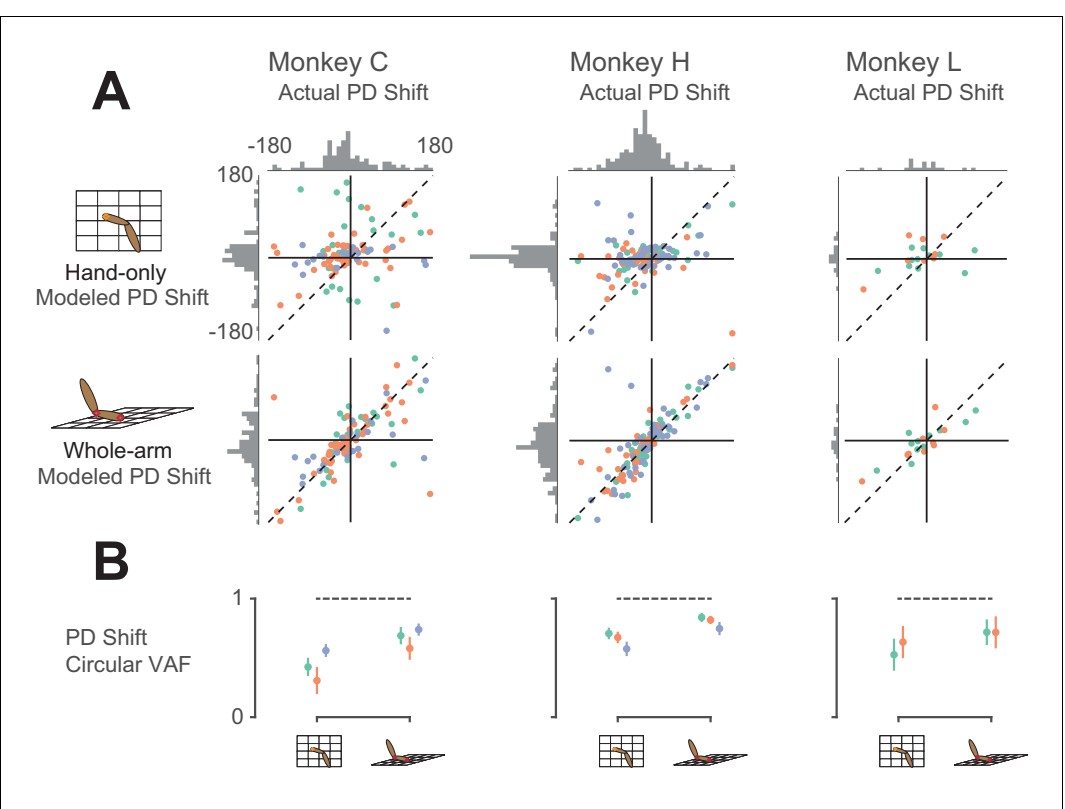

**Figure 7.** Model predictions of PD shift. (**A**) Scatter plots of model-predicted PD shifts plotted against actual PD shifts. Each dot represents the actual and modeled PD shifts of a single neuron, where different colors correspond to neurons recorded during different sessions. Dashed diagonal line shows perfect prediction. Horizontal histograms indicate distributions of actual PD shifts for each monkey. Vertical histograms indicate distributions of modeled shifts. Note that both horizontal and vertical axes are circular, meaning that opposing edges of the plots (top/bottom, left/right) are the same. Horizontal histograms show that the distribution of actual PD shifts included both clockwise and counter-clockwise shifts. Clustering of scatter plot points on the diagonal line for the whole-arm model indicates that it was more predictive of PD shift. (**B**) Plot showing circular VAF (cVAF) of scatter plots in A, an indicator of how well clustered points are around the diagonal line (see Materials and methods for details). Each point corresponds to the average cVAF for a model in a given session (indicated by color), and the horizontal dashed lines indicate the cVAF for perfect prediction. Error bars show 95% confidence intervals (derived from cross-validation – see Materials and methods). Pairwise comparisons between model cVAFs showed that the whole-arm model out-performed the hand-only model in all but one session, in which the two models were not significantly different.

counterintuitively, the actual changes included both clockwise and counter-clockwise rotations. However, we found that the whole-arm model predicted both types of PD changes quite well, indicated by a clustering of the scatter plot points in *Figure 7A* along the dashed diagonal line. Based on the circular VAF (cVAF; see Materials and methods for details) of the predicted PD changes, *Figure 7B* shows that the whole-arm model once again out-performed the hand-only model, with an average cVAF over all neurons of 0.75 compared to 0.57. We made pairwise comparisons between models for each session. In every session but one, the whole-arm model out-performed the hand-only model. In the remaining session, the difference between the two models was not significant ($p > 0.05$). These results lead to the same conclusion as the $pR^2$ and tuning curve correlation analyses: the kinematics of the whole-arm are important predictors of area 2 activity, and can explain differences between activity in the two workspaces that classic models cannot.

## Area 2 represents passive movements differently from active reaches

Given our success at modeling neural activity across workspaces with the whole-arm model, we set out to examine its effectiveness in a task that compared area 2 activity during active reaches and passive limb perturbations.

For this experiment, the monkey performed a center-out reaching task to four targets. On half of these trials, the monkey's hand was bumped by the manipulandum during the center-hold period in one of the four target directions (*Figure 8A*; see Materials and methods section for task details). This experiment included two sessions with each of Monkeys C and H. As in the earlier study performed by our group (*London and Miller, 2013*), we analyzed behavior and neural activity only during the 120 ms after movement onset for which the kinematics of the hand were similar in active and passive trials (*Figure 8B and C*). This is also the time period in which we can reasonably expect there not to be a voluntary reaction to the bumps in the passive trials.

Despite the similar hand kinematics in the active and passive movements, we found that whole-arm kinematics were quite different between the two conditions. Averaged over the sessions, a linear discriminant analysis (LDA) classifier could predict the movement type 89% of the time, using only the whole-arm kinematics in the analysis window, meaning that these whole-arm kinematics were highly separable based on movement condition. Considering our results from the two-workspace experiment, we would thus expect that the activity of area 2 neurons would also be highly separable.

As reported earlier, area 2 neurons had a wide range of sensitivities to active and passive hand movements (*London and Miller, 2013*). *Figure 8D* shows this difference for the neurons recorded during one session from Monkey C. As with our separability analysis for arm kinematics, we used LDA to classify movement type based on individual neurons, calling this prediction rate the neuron's 'separability index' (*Figure 8E*). We found that many neurons had an above chance separability index, as we would expect from neurons representing whole-arm kinematics.

There is thus a clear analogy between this experiment and the two-workspace experiment—both have two conditions which altered both the kinematics of the arm and the neural responses. Continuing the analogy, we asked how well our two models could predict neural activity across active/passive conditions. As with the two-workspace experiment, we fit both the hand-only and whole-arm models to neural activity during both active and passive movements, and found that the whole-arm model again tended to out-perform the hand-only model (Filled circles above the dashed unity line in *Figure 9*). However, there were many more neurons (open circles) for which the difference between models was insignificant compared to the two-workspace experiment (*Figure 4*).

As in the two-workspace experiment, we compared models trained within an individual (active or passive) condition, to those trained in both conditions (*Figure 10*). A number of neurons had consistent relationships with arm kinematics, indicated by the dots with positive $pR^2$ values lying close to the unity line. Surprisingly however, unlike our results from the two-workspace experiment (see *Figure 5*), many neurons in the active/passive task did not have this consistent relationship, indicated by the many neurons with negative $pR^2$ values for the model trained over both conditions.

The initial question of this experiment remains, however: does the neural separability index stem simply from arm kinematics? If this were true, then neurons with high separability index should have a consistent relationship to arm kinematics. To test this, we compared each neuron's $pR^2$ value when trained on both conditions (our proxy for model consistency) against its separability index (*Figure 11*). Interestingly, we found the opposite result—model consistency actually correlated

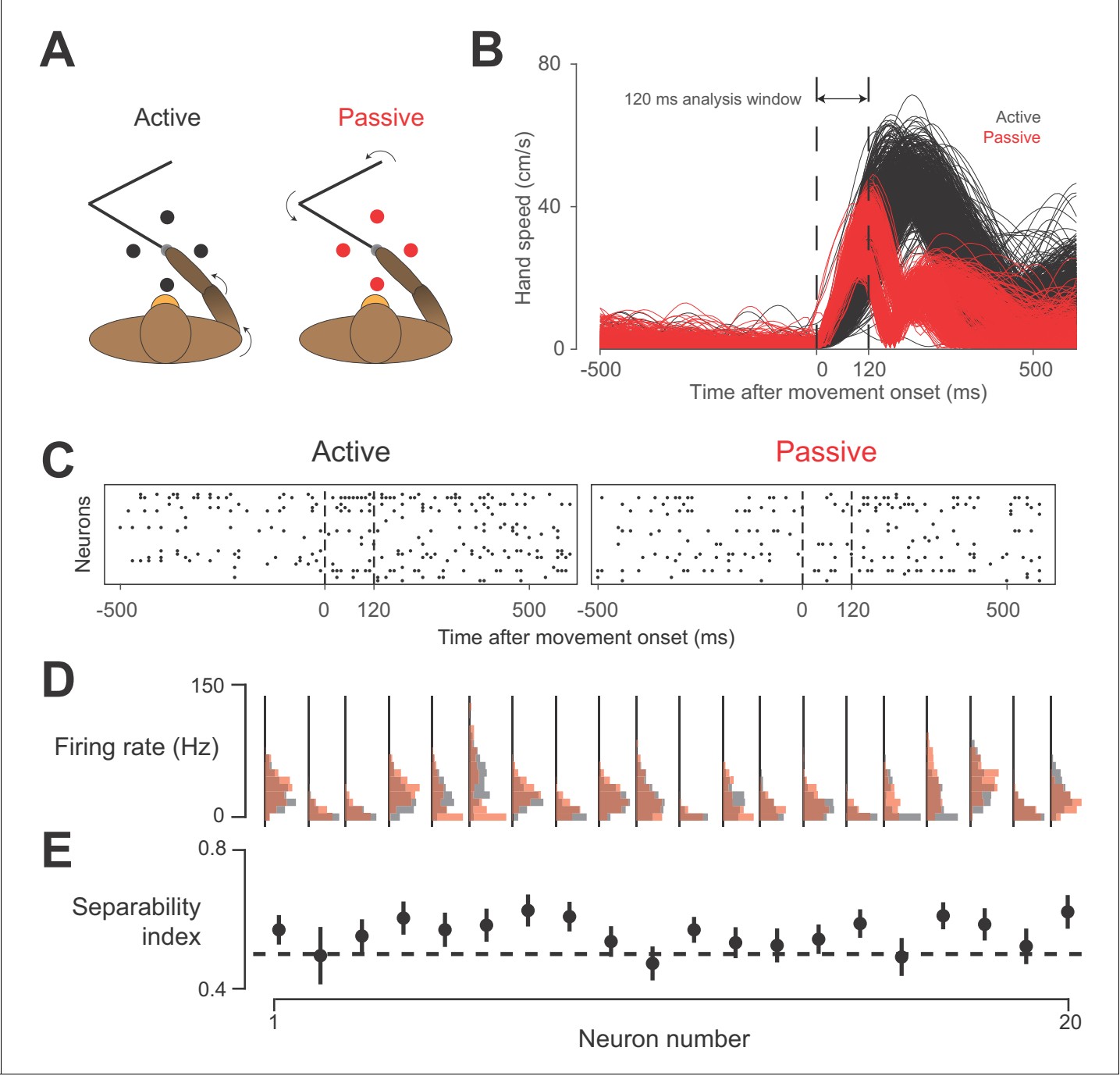

**Figure 8.** Active vs.passive behavior. (A) Schematic of task. On active trials (black), monkey reaches from center target to a target presented in one of four directions, indicated by the black circles. On passive trials, manipulandum bumps monkey's hand in one of the four target directions (red circles). (B) Speed of hand during active (black) and passive (red) trials, plotted against time, for one session. Starting around 120 ms after movement onset, a bimodal distribution in passive movement speed emerges. This bimodality reflects differences in the impedance of the arm for different directions of movement. Perturbations towards and away from the body tended to result in a shorter overall movement than those to the left or right. However, average movement speed was similar between active and passive trials in this 120 ms window, which we used for data analysis. (C) Neural raster plots for example active and passive trials for rightward movements. In each plot, rows indicate spikes recorded from different neurons, plotted against time. Vertical dashed lines delimit the analysis window. (D) Histograms of firing rates during active (black) and passive (red) movements for 20 example neurons from one session with Monkey H. (E) Separability index for each neuron during the session, found by testing how well linear discriminant analysis (LDA) could predict movement type from the neuron's average firing rate on a given trial. Black dashed line indicates chance level separability. Error bars indicate 95% confidence interval of separability index.

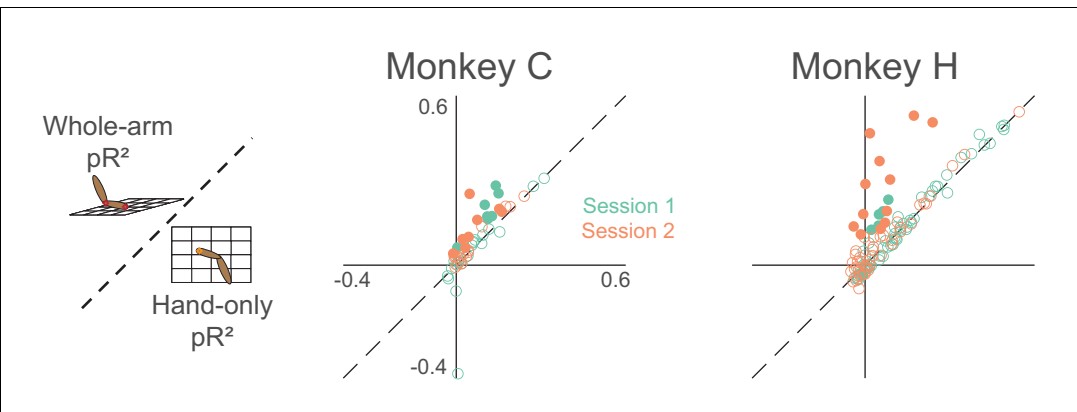

**Figure 9.** Goodness-of-fit comparison analysis for active/passive experiment (same format as *Figure 4*). Each dot represents a single neuron, with color indicating the recording session. Filled circles indicate neurons that are significantly far away from the dashed unity line.

negatively with the separability index. Essentially, this means that neurons responding to active and passive movements differently are likely not drawing this distinction based on arm kinematics, as those are the neurons for which we could not find a consistent whole-arm model. Instead, this suggests that neurons in area 2 distinguish active and passive movements by some other means, perhaps an efference copy signal from motor areas of the brain (*Bell, 1981*; *London and Miller, 2013*; *Nelson, 1987*).

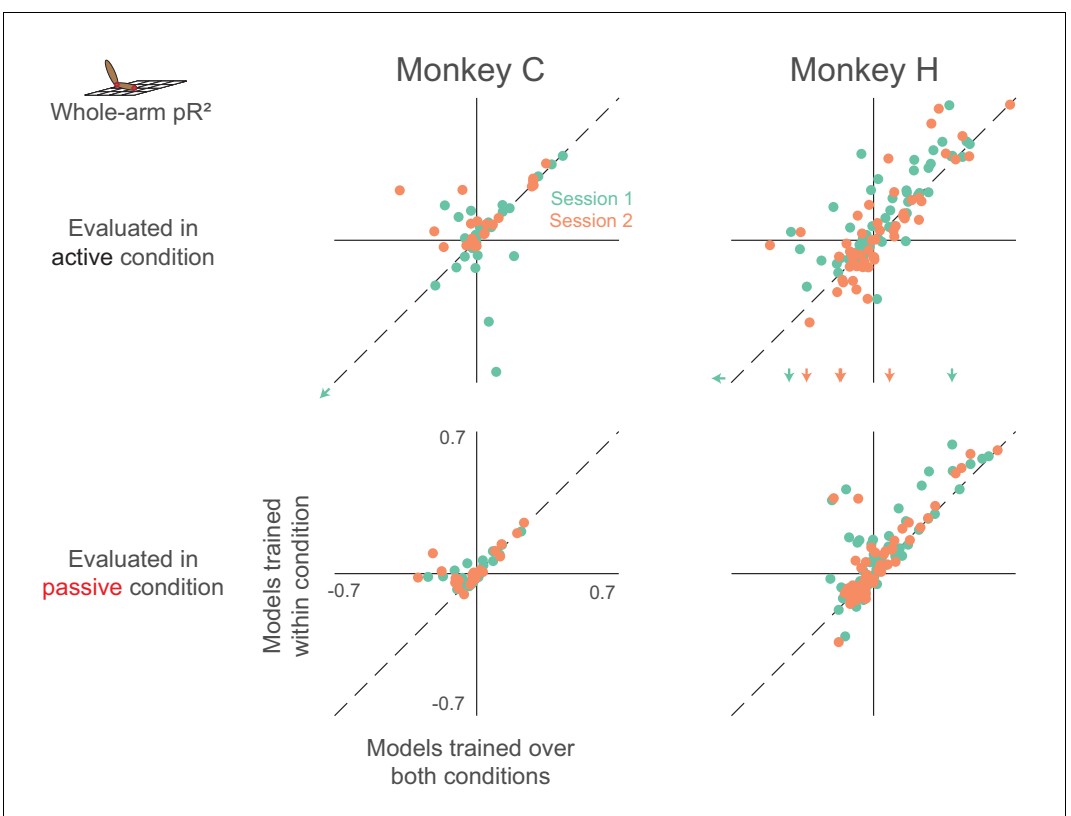

**Figure 10.** Dependence of whole-arm model accuracy on active and passive training data (same format as *Figure 5*). Plots in the upper row contain colored arrows at the edges indicating neurons with $pR^2$ value beyond the axis range, which we omitted for clarity.

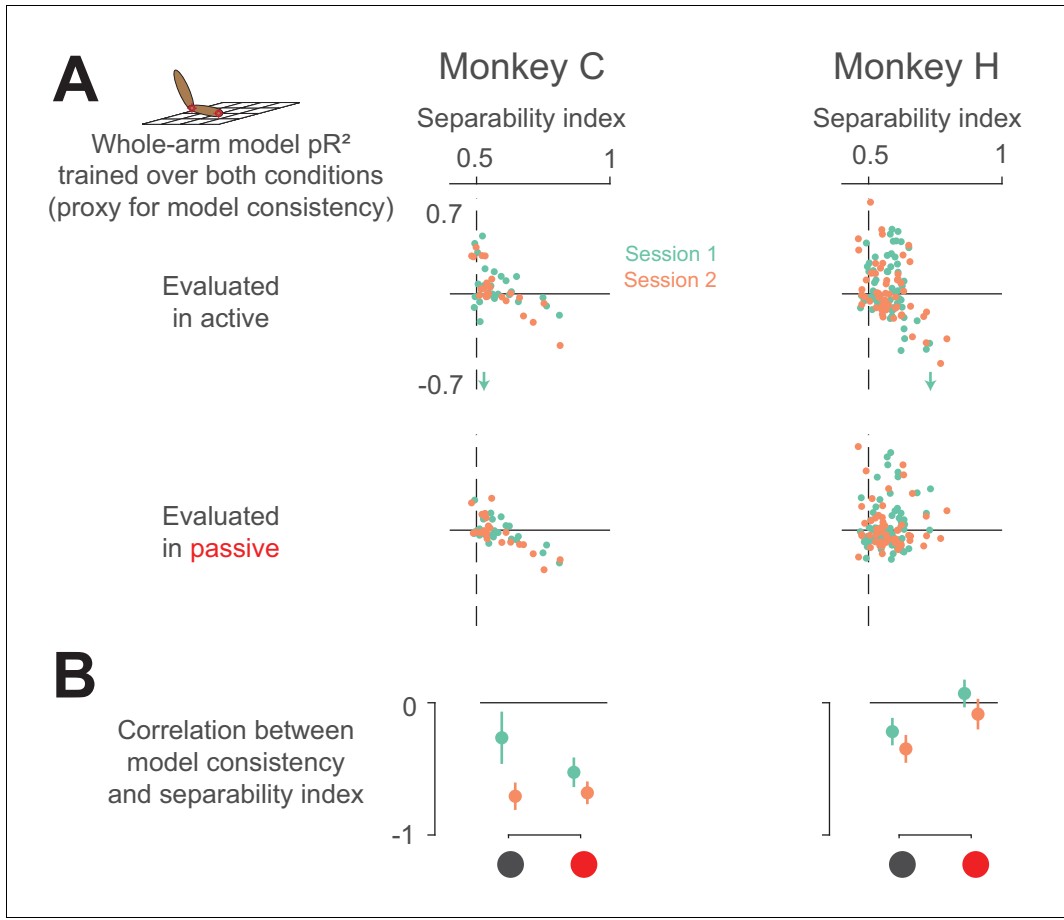

**Figure 11.** Neural separability index predicts whole-arm model inconsistency. (**A**) Scatter plots comparing the consistency of the whole-arm model against the separability index. Conventions are the same as in *Figure 10*. (**B**) Correlation between model consistency and separability index. Each dot represents the correlation between model consistency and separability index for a given session, with error bars representing the 95% confidence intervals.

## Discussion

### Summary

In this study, we explored, in two separate experiments, how somatosensory area 2 represents arm movements. In the first experiment, a monkey reached to targets in two separate workspaces. We found that a model incorporating whole-arm kinematics explained area 2 neural activity well, especially when compared to the hand-only model typically used to explain these neurons' responses. Our results from the experiment thus suggest that area 2 represents the state of the whole arm during reaching. In the second experiment, we tested the whole-arm model's ability to explain area 2 neural activity both during reaching, and when the hand was unexpectedly displaced passively. As in the first experiment, these two conditions differed both kinematically and in the neural responses to movement. However, we found that while some neurons maintained a consistent relationship with arm kinematics across the two conditions, many others did not. Furthermore, those neurons most sensitive to movement type were also those most poorly modeled across conditions. The results from this second experiment suggest that for some neurons, area 2 relates to arm kinematics differently for active and passive movements.

### Model complexity

A significant difference between the hand and whole-arm models is their number of parameters, which make the whole-arm models more complex and expressible. There are two concerns with

testing models of differing complexity, the first dealing with model training and evaluation, and the second with interpretation of the results.

In training and evaluating our models, we had to make sure that the complex models did not overfit the data, resulting in artificially high performance on the training dataset but low generalizability to new data. However, because we found through cross-validation that the more complex models generalized to test data better than the simpler models, they were not overfitting. Consequently, the hand-based models are clearly impoverished compared to the whole-arm models.

The second concern is in interpreting what it means when the more complex models perform better. One interpretation is that this is an obvious result; if the added degrees of freedom have anything at all to do with area 2 neural activity, then the more complex models should perform better. In fact, our main goal was primarily to improve our understanding of this area of S1 by exploring how incorporating measurements of whole-arm kinematics could help explain its function. As a result, we found that the whole-arm model not only out-performed the hand-only model, but it also predicted changes in PD across the two workspaces well in its own right. Furthermore, as demonstrated by the findings from our second experiment, the more complex model does not necessarily lead to a satisfactory fit. Despite its increased complexity and its success in the two-workspace task, the whole-arm model could not find a consistent fit for many neurons over both active and passive movements. As such, the active/passive experiment serves as a useful control for the two-workspace findings.

## Coordinate frame vs. informational content

Because of their differing dimensionality, the signals from the hand-only model and those from whole-arm model do not have a one-to-one relationship: there are many different arm configurations that result in a given hand position. Thus, a comparison between the hand-only and whole-arm models is mainly a question of information content (do area 2 neurons have information about more than just the hand?). In contrast, signals predicted by the various whole-arm models (see Appendix 1) do have a one-to-one (albeit nonlinear) relationship to each other. Knowledge of the hand and elbow position should completely determine estimated joint angles and musculotendon lengths, indicating that these models should have the same informational content. As such, a comparison between these models (as in the Appendix 1) is purely one of coordinate frame. While the interpretation for a comparison of information content is straightforward, interpreting the results of a comparison between coordinate frames is not. One major issue is that these comparisons only make sense when using linear models to relate neural activity to behavior. Once nonlinear models are considered, as in our study with artificial neural networks (*Lucas et al., 2019*), coordinate frames with one-to-one correspondence become nearly equivalent, and much more difficult to compare meaningfully.

Clear parallels exist between this and earlier studies seeking to find a unique representation of movement in motor areas. Over the last few decades, a controversy involving the exact nature of the neural representation of movement has played itself out in the literature surrounding motor cortex, with some advocating a hand-based representation of motor control (*Georgopoulos et al., 1982*; *Georgopoulos et al., 1986*; *Moran and Schwartz, 1999*) and others a muscle-based representation (*Evarts, 1968*; *Fetz et al., 1989*; *Morrow et al., 2007*; *Oby et al., 2013*). Recently, the motor control field started turning away from questions of coordinate frame and towards questions of neural population dynamics and information processing (*Churchland et al., 2010*; *Elsayed et al., 2016*; *Gallego et al., 2017*; *Kaufman et al., 2014*; *Perich et al., 2018*; *Russo et al., 2018*; *Sussillo et al., 2015*). Part of the motivation for this pivot in viewpoint is that it became increasingly clear that a 'pure' coordinate frame of movement representation is unlikely to exist (*Fetz, 1992*; *Kakei et al., 1999*). Further, studies tended to use correlation between neural activity and behavioral variables as evidence that the neurons represent movements in a particular coordinate frame. However, as noted above, these correlations could often be explained by multiple coordinate frames, casting doubt on the conclusiveness of the exact coordinate frame of representation (*Mussa-Ivaldi, 1988*). Consequently, in our study, we put aside the question of the coordinate frame of area 2, focusing instead on what we can gain by modeling area 2 in terms of whole-arm kinematics.

A major question this study leaves open is that of how information about reaching is processed by different areas of the proprioceptive neuraxis. While we might expect a muscle spindle-like representation at the level of the dorsal root ganglia (DRG) or the cuneate nucleus, downstream from the receptors by only one and two synapses, respectively, this representation likely changes as the

signals propagate through thalamus and into S1. Different areas of S1 may also have different representations. Area 3a, which receives input mostly from muscle afferents (*Heath et al., 1976*; *Kaas et al., 1979*; *Phillips et al., 1971*; *Yamada et al., 2016*), seems more likely to retain a muscle-like representation than is area 2, which integrates muscle afferent input with that from cutaneous receptors (*Hyvärinen and Poranen, 1978*; *Padberg et al., 2019*; *Pons et al., 1985*). Likewise, area 5 may have an even higher-level representation, as it receives input from both somatosensory (*Mountcastle et al., 1975*) and motor cortices (*Padberg et al., 2019*), and appears to depend on attention (*Chapman et al., 1984*; *Omrani et al., 2016*). As it becomes increasingly feasible to record from several of these areas simultaneously (*Richardson et al., 2016*; *Suresh et al., 2017*; *Weber et al., 2006*), future experiments could examine how these areas project information to each other, as has been explored in motor and premotor cortices (*Churchland et al., 2010*; *Elsayed et al., 2016*; *Kaufman et al., 2014*; *Perich et al., 2018*), without modeling the more complex cortical areas explicitly in terms of particular behavioral variables 'encoded' by single neurons.

## Possible evidence of efference copy in area 2

Our inability to find a consistent model across conditions suggests a difference between neural activity during active and passive movements that can't be captured by either model. One possible explanation for this is that area 2 may represent arm kinematics nonlinearly. Because we modeled area 2 activity with a generalized linear model (GLM; see Materials and methods), we implicitly discounted this possibility. The fact that the whole-arm kinematics for the two conditions are highly discriminable (89% separable on average) means that the different conditions correspond to different zones of kinematic space. Following the analogy of fitting a line to data distributed on an exponential curve, it is possible that the neurons with inconsistent linear relationships to arm kinematics may simply reflect a single nonlinear relationship, requiring different linear approximations in the two zones. Indeed, several of these neurons had high $pR^2$ for models trained within condition (top left quadrants of *Figure 10*).

Another possible explanation for this finding is that voluntary movements may change the afferent activity from the moving limb. This could be caused by altered descending gamma drive to muscle spindles that changes their sensitivity (*Loeb et al., 1985*; *Prochazka and Wand, 1981*; *Prochazka et al., 1976*). Another possibility is that of an efference copy signal sent to the brainstem or S1 from motor areas during active movements (*Bell, 1981*; *London and Miller, 2013*; *Nelson, 1987*). Many studies suggest that we use internal forward models of our bodies and the environment to coordinate our movements and predict their sensory consequences (*Shadmehr and Mussa-Ivaldi, 1994*; *Wolpert et al., 1995*). A key piece of this framework is comparing the actual feedback received following movement with the feedback predicted by the internal model, which generates a sensory prediction error. Recent studies suggest that S1 is important for updating the internal model using a sensory prediction error (*Mathis et al., 2017*; *Nasir et al., 2013*). Thus, one potential avenue to study the effect of efference copy in S1 would be to examine how motor areas communicate with area 2 during active and passive movements.

## Relevance for BCI

One motivation for this work is its potential to augment brain-computer interfaces (BCI) for restoring movement to persons with spinal cord injury or limb amputation. As BCI for motor control gets more advanced (*Collinger et al., 2013*; *Ethier et al., 2012*; *Kao et al., 2015*; *Young et al., 2019*), it will become more necessary to develop a method to provide feedback about movements to the brain, potentially using intracortical microstimulation (ICMS) to activate somatosensory areas. While the use of ICMS in S1 has led to some success in providing feedback about touch (*Flesher et al., 2016*; *Romo et al., 1998*; *Armenta Salas et al., 2018*; *Tabot et al., 2013*), the path towards providing proprioceptive feedback remains relatively unexplored. At least one study did use electrical stimulation in S1 for feedback during movement, using the stimulation to specify target direction with respect to the evolving hand position (*Dadarlat et al., 2015*). However, this target-error information is very different from the information normally encoded by S1, and the monkeys required several months to learn to use it. To our knowledge, no study has yet shown a way to use ICMS to provide more biomimetic proprioceptive feedback during reaching.

Previously, our lab attempted to address this gap by stimulating electrodes in area 2 with known neural PDs. In one monkey, ICMS delivered simultaneously with a mechanical bump to the arm biased the monkey's perception of the bump direction toward the electrodes' PD (*Tomlinson and Miller, 2016*). Unfortunately, the result could not be replicated in other monkeys; while the ICMS often biased their reports, the direction of the bias could not be explained by the PDs of the stimulated electrodes. One potential reason may be that the stimulation paradigm in those experiments was derived from the classic, hand-based model and the assumption that area 2 represents active and passive movements similarly. As this paper has shown, both of these assumptions have important caveats. It is possible that a stimulation paradigm based on a whole-arm model may be more successful.

Another important consideration is the difference between sensation for perception versus action, which which is thought to arise from processing in two distinct pathways (*Dijkerman and de Haan, 2007*; *Mishkin and Ungerleider, 1982*; *Sedda and Scarpina, 2012*). Most studies using ICMS have tended to engage the perceptual rather than the action stream of proprioception, either by using perceptual reporting of the effects of ICMS (*Armenta Salas et al., 2018*; *Tomlinson and Miller, 2016*; *Zaaimi et al., 2013*), or by using ICMS as a conscious sensory substitute for prorioceptive feedback (*Dadarlat et al., 2015*). However, as we better characterize how S1 represents movements, we hope to forge a way towards a stimulation paradigm in which we can engage both streams, to enable users of a BCI both to perceive their limb, and to respond rapidly to movement perturbations.

## Conclusion

Our goal in conducting this study was to improve our understanding of how area 2 neural activity represents arm movements. We began by asking what we would learn about area 2 when we tracked the movement of the whole arm, rather than just the hand. The results of our first experiment showed that a model built on these whole-arm kinematics was highly predictive of area 2 neural activity, suggesting that it indeed represents the kinematic state of the whole arm during reaching. In our second experiment, we sought to extend these findings to similar movements when the limb is passively displaced. There, we found that while some neurons consistently represented arm kinematics, others did not, suggesting that the area may process active and passive movements differently, possibly with the addition of efference copy inputs.

# Materials and methods

**Key resources table**

| Reagent type (species) or resource | Designation | Source or reference | Identifiers | Additional information |
|---|---|---|---|---|
| Software, algorithm | MATLAB | MathWorks | RRID: SCR_001622 | All code developed for this paper available on GitHub (See relevant sections of Materials and methods) |

All surgical and experimental procedures were fully consistent with the guide for the care and use of laboratory animals and approved by the institutional animal care and use committee of Northwestern University under protocol #IS00000367.

## Behavior

We recorded data from a monkey while it used a manipulandum to reach for targets presented on a screen within a 20 cm x 20 cm workspace. After each successful reaching trial, the monkey received a pulse of juice or water as a reward. We recorded the position of the handle using encoders on the manipulandum joints. We also recorded the interaction forces between the monkey's hand and the handle using a six-axis load cell mounted underneath the handle.

For the two-workspace experiment, we partitioned the full workspace into four 10 cm x 10 cm quadrants. Of these four quadrants, we chose the far ipsilateral one and the near contralateral one in which to compare neural representations of movement. Before each trial, we chose one of the two workspaces randomly, within which the monkey reached to a short sequence of targets randomly positioned in the workspace. For this experiment, we only analyzed the portion of data from the end of the center-hold period to the end of the trial.

For the active vs. passive experiment, we had the monkey perform a classic center-out (CO) reaching task, as described in *London and Miller (2013)*. Briefly, the monkey held in a target at the center of the full workspace for a random amount of time, after which one of four outer targets was presented. The trial ended in success once the monkey reached to the outer target. On 50% of the trials (deemed 'passive' trials), during the center hold period, we used motors on the manipulandum to deliver a 2 N perturbation to the monkey's hand in one of the four target directions. After the bump, the monkey returned to the center target, after which the trial proceeded like an active trial. From only the successful passive and active trials, we analyzed the first 120 ms after movement onset. Movement onset was determined by looking for the peak in handle acceleration either after the motor pulse (in the passive condition) or after 200 ms post-go cue (in the active condition) and sweeping backwards in time until the acceleration was less than 10% of the peak.

## Motion tracking

Before each reaching experiment, we painted 10 markers of four different colors on the outside of the monkey's arm, marking bony landmarks and a few points in between, a la (*Chan and Moran, 2006*). Using a custom motion tracking system built from a Microsoft Kinect, we recorded the 3D locations of these markers with respect to the camera, synced in time to the other behavioral and neural data. We then aligned the Kinect-measured marker locations to the spatial lab frame by aligning location of the Kinect hand marker to the location of the handle in the manipulandum coordinate frame. Code for motion tracking can be found at https://github.com/limblab/KinectTracking.git (*Chowdhury, 2020a*; copy archived at https://github.com/elifesciences-publications/KinectTracking).

## Neural recordings

We implanted 100-electrode arrays (Blackrock Microsystems) into the arm representation of area 2 of S1 in these monkeys. For more details on surgical techniques, see *Weber et al. (2011)*. In surgery, we roughly mapped the postcentral gyrus by recording from the cortical surface while manipulating the arm and hand to localize their representations. To record neural data for our experiments, we used a Cerebus recording system (Blackrock). This recording system sampled signals from each of the 96 electrodes at 30 kHz. To conserve data storage space, the system detected spikes online using a threshold set at $-5x$ signal RMS, and only wrote to disk a time stamp and the 1.6 ms snippet of signal surrounding the threshold crossing. After data collection, we used Plexon Offline Sorter to manually sort these snippets into putative single units, using features like waveform shape and interspike interval.

## Sensory mappings

In addition to recording sessions, we also occasionally performed sensory mapping sessions to identify the neural receptive fields. For each electrode we tested, we routed the corresponding recording channel to a speaker and listened to multi-unit neural activity while manipulating the monkey's arm. We noted both the modality (deep or cutaneous) and the location of the receptive field (torso, shoulder, humerus, elbow, forearm, wrist, hand, or arm in general). We classified an electrode as cutaneous if we found an area of the skin, which when brushed or stretched, resulted in an increase in multi-unit activity. We classified an electrode as deep if we found activity to be responsive to joint movements and/or muscle palpation but could not find a cutaneous field. As neurons on the same electrode tend to have similar properties (*Weber et al., 2011*), we usually did not separate neurons on individual electrodes during mapping. However, when we did, we usually found them to have similar receptive field modality and location.

In Monkeys C and H, we found a gradient of receptive field location across the array, corresponding to a somatotopy from proximal to distal. To quantify this gradient, we assigned each receptive

field location a score from 1 to 7 (with 1 being the torso and 7 being the hand), and we fit a simple linear model relating this location on the limb to the x and y coordinates of electrodes on the array. We show the calculated gradients for Monkeys C and H as black arrows in *Figure 1* (both significant linear fits with p < 0.05). Monkey L's array had too few neurons to calculate a significant linear model.

## Neural analysis

Code for the following neural analyses can be found at https://github.com/raeedcho/s1-kinematics. git (*Chowdhury, 2020b*; copy archived at https://github.com/elifesciences-publications/s1-kinematics).

### Preferred directions

We used a simple bootstrapping procedure to calculate PDs for each neuron. On each bootstrap iteration, we randomly drew timepoints from the reaching data, making sure that the distribution of movement directions was uniform to mitigate the effects of any potential bias. Then, as in *Georgopoulos et al. (1982)*, we fit a cosine tuning function to the neural activity with respect to the movement direction, using *Equation 1a, b*.

$$f_i(\tau) = b_0 + b_1 * \sin(\theta_m(\tau)) + b_2 * \cos(\theta_m(\tau)) \tag{1a}$$

$$= b_0 + r_i * \cos(\theta_m(\tau) - PD_i) \tag{1b}$$

where

$$PD_i = atan2(b_1, b_2) \text{ and } r_i = \sqrt{b_1^2 + b_2^2}$$

Here, $f_i(\tau)$ is the average firing rate of neuron $i$ for a given time point $\tau$, and $\theta_m(\tau)$ is the corresponding movement direction, which for the active/passive task was the target or bump direction, and for the two-workspace experiment was the average movement direction over a time bin. We took the circular mean of $PD_i$ and mean of $r_i$ over all bootstrap iterations to determine the preferred direction and the modulation depth respectively, for each neuron.

As the PD analysis is meaningless for neurons that don't have a preferred direction of movement, we only analyzed the PDs of neurons that were significantly tuned. We assessed tuning through a separate bootstrapping procedure, described in *Dekleva et al. (2018)*. Briefly, we randomly sampled the timepoints from reaching data, again ensuring a uniform distribution of movement directions, but this time also randomly shuffled the corresponding neural activity. We calculated the $r_i$ for this shuffled data on each bootstrap iteration, thereby creating a null distribution of modulation depths. We considered a neuron to be tuned if the true $r_i$ was greater than the 95th percentile of the null distribution.

### Models of neural activity

For the two-workspace analyses, both behavioral variables and neural firing rate were averaged over 50 ms bins. For the active/passive analyses, we averaged behavioral variables and neural firing rates over the 120 ms period following movement onset in each trial. We modeled neural activity with respect to the behavior using Poisson generalized linear models (outline in *Truccolo et al., 2005*) shown in *Equation 2a*, below.

$$f \sim Poisson(\lambda), \ \lambda = \exp(X\beta) \tag{2a}$$

In this equation, $f$ is a $T$ (number of time points) x $N$ (number of neurons) matrix of average firing rates, $X$ is a $T$ x $P$ (number of behavioral covariates, explained below) matrix of behavioral correlates, and $\beta$ is a $P$ x $N$ matrix of model parameters. We fit these GLMs by finding maximum likelihood estimation of the parameters, $\hat{\beta}$. With these fitted models, we predicted firing rates ($\hat{f}$) on data not used for training, shown in *Equation 2b*, below.

$$\hat{f} = \exp(X\hat{\beta}) \tag{2b}$$

We tested six firing rate encoding models, detailed below. Of these six models, the first two (hand-only and whole-arm) were the ones shown in the main text, with results from the other models detailed in Appendix 1. Note that each model also includes an offset term, increasing the number of parameters, $P$, by one.

- Hand-only: behavioral covariates were position and velocity of the hand, estimated by using the location of one of the hand markers, in three-dimensional Cartesian space, with origin at the shoulder ($P = 7$).
- Whole-arm: behavior covariates were position and velocity of both the hand and elbow markers in three-dimensional Cartesian space, with origin at the shoulder. This is the simplest extension of the extrinsic model that incorporates information about the configuration of the whole arm ($P = 13$).
- Hand kinematics+force: behavioral covariates were position and velocity of the hand, as well as forces and torques on the manipulandum handle, in three-dimensional Cartesian space ($P = 13$).
- Egocentric: behavior covariates were position and velocity of the hand marker in spherical coordinates ($\theta$, $\phi$, and $\rho$), with origin at the shoulder ($P = 7$).
- Joint kinematics: behavioral covariates were the 7 joint angles (shoulder flexion/abduction/rotation, elbow flexion, wrist flexion/deviation/pronation) and corresponding joint angular velocities ($P = 15$).
- Muscle kinematics: behavioral covariates were derived from the length of the 39 modeled muscles (*Chan and Moran, 2006*) and their time derivatives. However, because this would result in almost 78 (highly correlated) covariates, we used PCA to extract 5-dimensional orthogonal basis sets for both the lengths and their derivatives. On average, five components explained 99 and 96 percent of the total variance of lengths and length derivatives, respectively. Behavioral covariates of this model were the projections of the muscle variables into these spaces during behavior ($P = 11$).

We used repeated 5-fold cross-validation to evaluate our models of neural activity, given that the models had different numbers of parameters, $P$. On each repeat, we randomly split trials into five groups (folds) and trained the models on four of them. We used these trained models to predict neural firing rates ($f_i$) in the fifth fold. We then compared the predicted firing rates from each model to the actual firing rates in that test fold, using analyses described in the following sections. This process (including random splitting) was repeated 20 times, resulting in n=100 sample size for each analysis result. Thus, if a more expressive model with more parameters performs better than a simpler model, it would suggest that the extra parameters do provide relevant information about the neural activity not accounted for by the simpler models.

## Statistical tests and confidence intervals

To perform statistical tests on the output of repeated 5-fold cross-validation, we used a corrected resampled t-test, outlined in *Ernst (2017)* and *Nadeau and Bengio (2003)*. Here, sample mean and variance are calculated as in a normal t-test, but a correction factor needs to be applied to the standard error, depending on the nature of the cross-validation. *Equation 3a-c* shows a general case of this correction for R repeats of K-fold cross-validation of some analysis result $d_{kr}$.

$$\hat{\mu}_d = \frac{1}{K \times R} \sum_{k=1}^{K} \sum_{r=1}^{R} d_{kr} \tag{3a}$$

$$\hat{\sigma}_d^2 = \frac{1}{(K \times R) - 1} \sum_{k=1}^{K} \sum_{r=1}^{R} (d_{kr} - \hat{\mu}_d)^2 \tag{3b}$$

$$t_{stat} = \frac{\hat{\mu}_d}{\sqrt{\left(\frac{1}{K \times R} + \frac{1/K}{1 - 1/K}\right) \hat{\sigma}_d^2}} \tag{3c}$$

We then compare the t-statistic here ($t_{stat}$) to a t-distribution with $K \times R - 1$ degrees of freedom. The correction applied is an extra term (i.e., $\frac{1/K}{1-1/K}$) under the square root, compared to the typical

standard error calculation. Note that we performed all statistical tests within individual sessions or for individual neurons, never across sessions or monkeys.

## Bonferroni corrections

At the beginning of this project, we set out to compare three of these six models: hand-only, ego-centric, and muscle kinematics. In making pairwise comparisons between these models, we used $\alpha = 0.05$ and a Bonferroni correction of 3, for the three original comparisons. In this analysis, we found that the muscle model performed best. As we developed this project, however, we tried the three other models to see if they could outperform the muscle kinematics model, eventually finding that the whole-arm model, built on Cartesian kinematics of the hand and elbow outperformed it. As this appeared to be primarily due to modeling and measurement error in the muscle model (see Appendix 1), we decided to focus on the hand-only and whole-arm model. Despite only making one pairwise comparison in the main text, we chose to use a Bonferroni correction factor of 6: three for the original three pairwise comparisons and one more for each additional model we tested, which were compared against the best model at the time, and could have changed the end result of this project.

## Goodness-of-fit (pseudo-R$^2$)

We evaluated goodness-of-fit of these models for each neuron by using a pseudo-R$^2$ ($pR^2$) metric. We used a formulation of pseudo-R$^2$ based on a comparison between the deviance of the full model and the deviance of a 'null' model, that is, a model that only predicts the overall mean firing rate (**Cameron and Windmeijer, 1997**; **Cameron and Windmeijer, 1996**; **Heinzl and Mittlböck, 2003**; **Perich et al., 2018**).

$$pR^2 = 1 - \frac{D(f_i;\hat{f}_l)}{D(f_i;\bar{f}_i)} \tag{4a}$$

$$= 1 - \frac{logL(f_i) - logL(\hat{f}_l)}{logL(f_i) - logL(\bar{f}_i)} \tag{4b}$$

When computing the likelihood of a Poisson statistic, this is:

$$= 1 - \frac{\sum_{\tau=1}^{T} f_i(\tau) \log\left(\frac{f_i(\tau)}{\hat{f}_i(\tau)}\right) - (f_i(\tau) - \hat{f}_i(\tau))}{\sum_{\tau=1}^{T} f_i(\tau) \log\left(\frac{f_i(\tau)}{\bar{f}_i}\right) - (f_i(\tau) - \bar{f}_i)} \tag{4c}$$

This pR$^2$ metric ranges from $-\infty$ to 1, with a value of 1 corresponding to a perfectly fit model and a value of 0 corresponding to a model that only fits as well as the 'null' model. In contrast with the general intuition for regular R$^2$, a pR$^2$ of ~0.2 is considered a 'good' fit (**McFadden, 1977**).

## Tuning curves

We binned the trajectory into 16 bins, each 22.5 degrees wide, based on the mean direction across 50 ms of hand motion. For each directional bin, we calculated the sample mean and 95% confidence interval of the mean. In figures, we plotted this mean firing rate against the center-point of the bin.

## Preferred direction shift

We calculated PDs for each neuron in each workspace and found the predicted change in PD from the contralateral workspace to the ipsilateral workspace, given each model. We compared these changes to those observed for each neuron. The values of these PD shifts are shown in *Figure 7* for all neurons tuned to movements in both workspaces, averaged over all 100 test folds.

We computed a variance-accounted-for (VAF) metric, here called the 'circular VAF' (cVAF) for each neuron ($i$) in each fold as:

$$cVAF_i = cos\left(\Delta\theta_{PD,i} - \Delta\hat{\theta}_{PD,i}\right) \tag{5}$$

As the cVAF metric is essentially the inner product of unit vectors with direction $\Delta\theta_{PD,i}$ and $\Delta\hat{\theta}_{PD,i}$, it accounts for the circular domain of the PD shifts. Like regular VAF, the cVAF has a maximum value of 1 when $\Delta\theta_{PD,i}$ and $\Delta\hat{\theta}_{PD,i}$ are the same, and decreases in proportion to the squared difference between $\Delta\theta_{PD,i}$ and $\Delta\hat{\theta}_{PD,i}$. We took the average cVAF over all neurons as the cVAF for the fold. In total, given the 20 repeats of 5-fold cross-validation, this gave us 100-samples of the cVAF for each model in a given session.

### Separability index

In the active/passive experiment, we calculated the separability index for each neuron by fitting a linear discriminant analysis (LDA) classifier, predicting trial type (active or passive) from the neuron's average activity in the 120 ms after movement onset. As with the other neural analyses, we fit and evaluated each LDA classifier using our repeated 5-fold cross-validation scheme, calling the average test set classification percentage the neuron's separability index.

Our procedure for calculating the separability of the whole-arm kinematics was similar, simply substituting the whole-arm kinematics for the neural activity when training and testing the LDA classifier.

## Acknowledgements

We would like to thank Brian London for initial discussions of the active vs. passive result and Tucker Tomlinson, Christopher VerSteeg, and Joseph Sombeck for their help with training and caring for the research animals. Additionally, we would like to thank them, along with Matt Perich, Juan Gallego, Sara Solla, and the entire Miller Limb Lab for discussions and feedback that greatly improved this work.

This research was funded by National Institute of Neurological Disorders and Stroke Grant No. NS095251 and National Science Foundation Grant No. DGE-1324585.

## Additional information

### Funding

| Funder | Grant reference number | Author |
| --- | --- | --- |
| National Science Foundation | Graduate Research Fellowship DGE-1324585 | Raeed H Chowdhury |
| National Institutes of Health | R01 NS095251 | Lee E Miller |

The funders had no role in study design, data collection and interpretation, or the decision to submit the work for publication.

### Author contributions

Raeed H Chowdhury, Conceptualization, Data curation, Software, Formal analysis, Funding acquisition, Validation, Investigation, Visualization, Methodology, Project administration; Joshua I Glaser, Software, Formal analysis, Investigation, Methodology; Lee E Miller, Conceptualization, Resources, Supervision, Funding acquisition, Investigation, Methodology, Project administration

### Author ORCIDs

Raeed H Chowdhury ⓘ https://orcid.org/0000-0002-5934-919X
Joshua I Glaser ⓘ http://orcid.org/0000-0003-2906-115X
Lee E Miller ⓘ https://orcid.org/0000-0001-8675-7140

### Ethics

Animal experimentation: All surgical and experimental procedures were fully consistent with the guide for the care and use of laboratory animals and approved by the institutional animal care and use committee of Northwestern University under protocol #IS00000367.

**Decision letter and Author response**
Decision letter https://doi.org/10.7554/eLife.48198.sa1
Author response https://doi.org/10.7554/eLife.48198.sa2

## Additional files

### Supplementary files
• Transparent reporting form

### Data availability
All data used for this paper are posted on Dryad.

The following dataset was generated:

| Author(s) | Year | Dataset title | Dataset URL | Database and Identifier |
|---|---|---|---|---|
| Raeed H Chowdhury, Joshua I Glaser, Lee E Miller | 2020 | Data from: Area 2 of primary somatosensory cortex encodes kinematics of the whole arm | https://doi.org/10.5061/dryad.nk98sf7q7 | Dryad Digital Repository, 10.5061/dryad.nk98sf7q7 |

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

# Appendix 1

## Within class model comparisons

Over the course of this project, we analyzed several different models of area 2 activity. We categorized these models into two classes based on whether they contained information about the hand or the arm in different coordinate frames. Of these models, we picked the hand-only and whole-arm models to represent the two model classes in the main paper, as we found that the other within-class models offered little additional insight into area 2 activity. For completeness, however, this section expands on the comparisons between within-class models.

## Hand model comparison

Two of our models used the kinematics of hand movement as behavioral covariates for area 2 neural activity: the hand-only model in the main paper and the egocentric model, which represents hand kinematics in a spherical coordinate frame with origin at the shoulder. While the egocentric model, or a model like it, has been proposed as a possible coordinate frame for representation of the limb (*Bosco et al., 1996*; *Caminiti et al., 1990*), we found that it performed rather poorly at explaining neural activity in area 2 from the two-workspace task. *Appendix 1—figure 1A and B* show comparisons between the hand-only model and the egocentric model in terms of pR$^2$ and tuning curve correlation, as in the main paper. These comparisons show that the hand-only model tended to out-perform the egocentric model. Further, the egocentric model predicted large shifts in PD between the two workspaces (*Appendix 1—figure 1C*) that did not match up at all to the actual PD shifts.

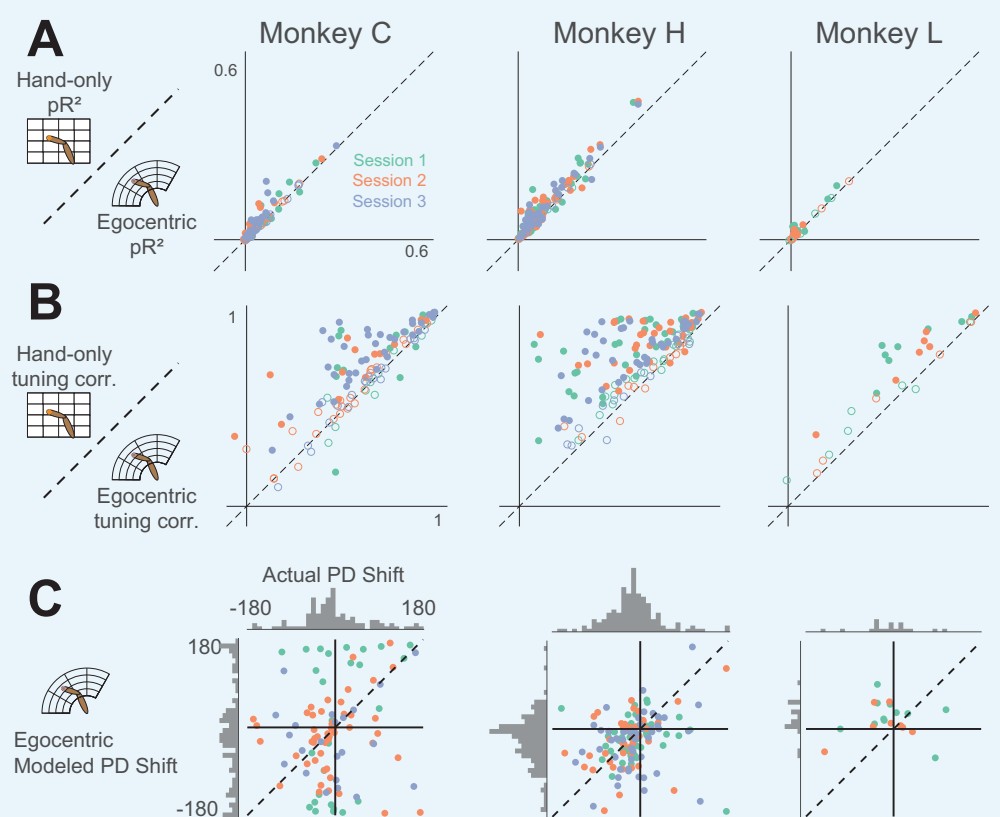

**Appendix 1—figure 1.** Comparison between hand-only model and egocentric model. (**A**) pR$^2$ comparison, as in *Figure 4*. (**B**) Tuning curve correlation comparison, as in *Figure 6*. (**C**) Modeled PD shift compared to actual PD shift for egocentric model, as in *Figure 7A*.

## Arm model comparison

In addition to the whole-arm model detailed in the main paper, we tested two models of area 2 activity based on biomechanics: one based on joint kinematics and the other based on musculotendon lengths. To find these behavioral covariates, we registered these marker locations to a monkey arm musculoskeletal model in OpenSim (SimTK), based on a model of the macaque arm published by *Chan and Moran (2006)*, and which can be found at https://github.com/limblab/monkeyArmModel.git (*Chowdhury, 2020c*; copy archived at https://github.com/elifesciences-publications/monkeyArmModel). After scaling the limb segments of the model to match those of each monkey, we used the inverse kinematics analysis tool provided by OpenSim to estimate the joint angles (and corresponding muscle lengths) required to match the model's virtual marker positions to the positions of the actual recorded markers. Previously, Chan and Moran used this model to analyze the joint and muscle kinematics as a monkey performs a center out task (*Chan and Moran, 2006*). Here, we use the musculoskeletal model to predict neural activity.

*Appendix 1—figure 2A and B* show comparisons of $pR^2$ and tuning curve correlation between the whole-arm model detailed in the paper and these two biomechanical models. We found that the three models provided similar predictions, but surprisingly, the whole-arm model generally outperformed the biomechanical models. *Appendix 1—figure 2C* shows the predicted PD shifts from these models, as in *Figure 7A*. We found that neither biomechanical model could predict PD shifts as well as the whole-arm model, though the muscle model in particular appeared to perform well.

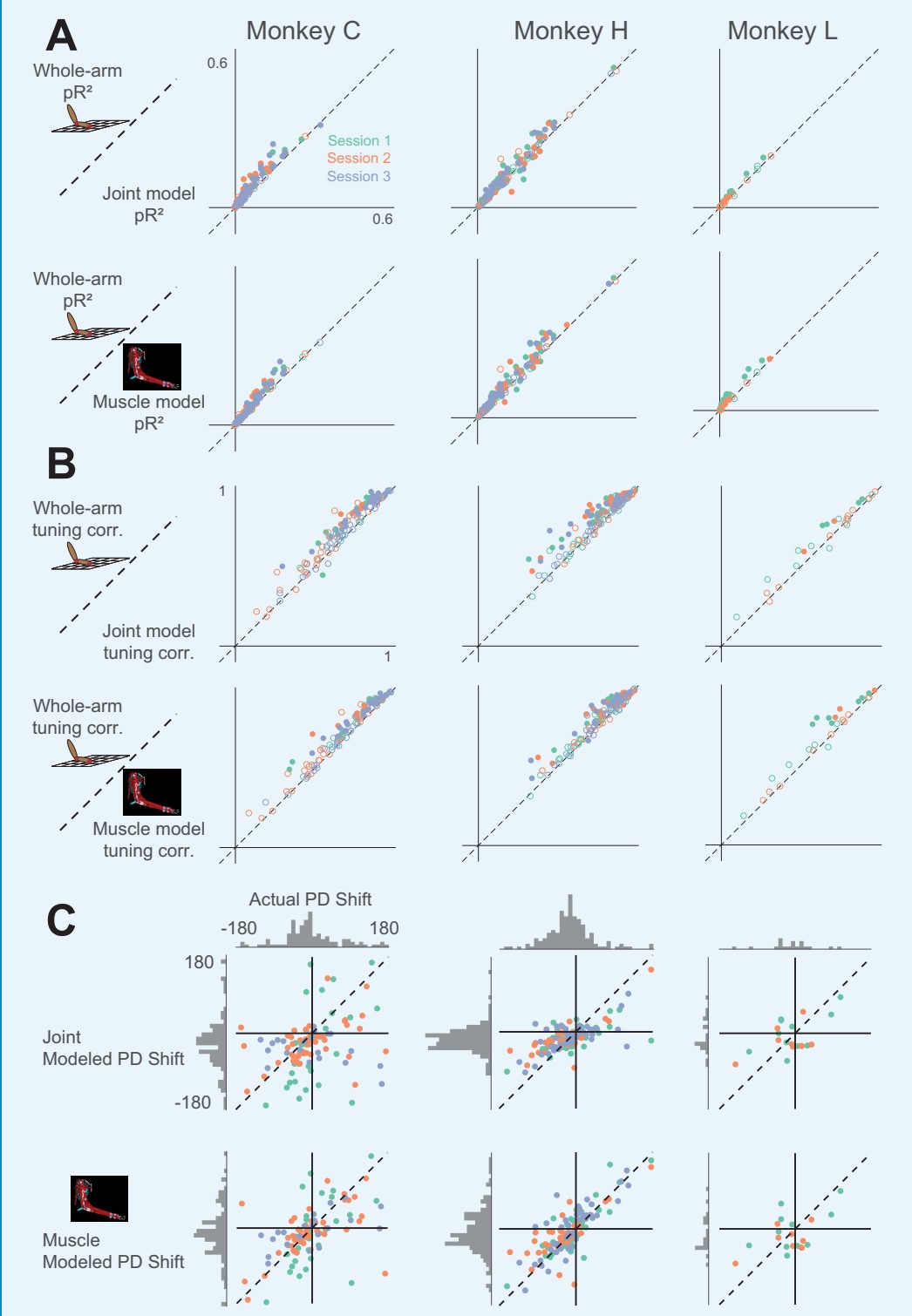

**Appendix 1—figure 2.** Comparison between whole-arm model and biomechanical models (joint kinematics and musculotendon length). Same arrangement as in *Appendix 1—figure 1*.

As a control for errors introduced into the muscle model by processing marker data with OpenSim, we performed the cVAF analysis on a whole-arm model where hand and elbow kinematics were derived from joint angles of the musculoskeletal model, rather than directly from the marker locations captured by the motion tracking system. We re-ran the model prediction analysis for only the muscle model, marker-derived whole-arm model, and

OpenSim-based whole-arm model. Unsurprisingly, we found average cVAFs similar to those from the main analysis for the marker-derived whole-arm model (0.75). However, the cVAF ofthe OpenSim-based whole-arm model (0.67) dropped to that ofthe muscle model (0.67). This suggests that the difference in predictive capability between the muscle and whole-arm models stems at least in part from errors introduced in OpenSim modeling, rather than from the whole-arm model necessarily being the better model for area 2 neural activity.

## Discussion of arm model comparisons

As proprioceptive signals originate in the muscles, arising from muscle spindles and Golgi tendon organs, we expected to find that the muscle model would outperform the other models. However, there are several potential reasons why this was not so. The most important ones can be divided into two categories loosely tied to 1) errors in estimating the musclulotendon lengths, through motion tracking and musculoskeletal modeling, and 2) the fidelity of the muscle model to the actual signals sent by the proprioceptors.

In the first category, the main issue is that of error propagation. The extra stages of analysis required to compute musculotendon lengths (registering markers to a musculoskeletal model, performing inverse kinematics to find joint angles, and using modeled moment arms to estimate musculotendon lengths) introduce errors not present when simply using the positions of markers on the arm. As a control, we ran the whole-arm model through two of these extra steps by computing the hand and elbow positions from the joint angles of the scaled model, estimated from inverse kinematics. The results of this analysis showed that the performance of the whole-arm model with added noise dropped to that of the muscle model, indicating that there are, in fact, errors introduced in even this portion of the processing chain.

The other potential source of error in this processing chain stems from the modeled moment arms, which might not accurately reflect those of the actual muscles. In developing their musculoskeletal model, Chan and Moran collected muscle origin and insertion point measurements from both cadaveric studies and existing literature (*Chan and Moran, 2006*). However, due to the complexity of some joints, along with ambiguity of how the muscle wraps around bones and other surfaces, determining moment arms purely by bone and muscle geometry is a difficult problem (*An et al., 1984*). Because moment arms are irrelevant for determining hand and elbow kinematics, we could not subject the whole-arm model to the error introduced by this step.

In addition to the questions of error propagation and musculoskeletal model accuracy is the question of whether our muscle model was truly representative of the signals sensed by the proprioceptors. The central complication is that spindles sense the state of the intrafusal fibers in which they reside, and have a complex, nonlinear relation to the musculotendon length that we used in our muscle model. Factors like load-dependent fiber pennation angle (*Azizi et al., 2008*), or tendon elasticity (*Rack and Westbury, 1984*) can decouple muscle fiber length from musculotendon length. Additionally, intrafusal fibers receive motor drive from gamma motor neurons, which continuously alters muscle spindle sensitivity (*Loeb et al., 1985*; *Prochazka and Wand, 1981*; *Prochazka et al., 1976*) and spindle activity also depends on the history of strain on the fibers (*Haftel et al., 2004*; *Proske and Stuart, 1985*). Altogether, this means that while the musculotendon lengths we computed provide a reasonably good approximation of what the arm is doing, they may not be a good representation of the spindle responses themselves. Spindle activity might be more accurately modeled when given enough information about the musculotendon physiology. However, to model the effects of gamma drive, we would either have to record directly from gamma motor neurons or make assumptions of how gamma drive changes over the course of reaching. In developing models of neural activity, one must carefully consider the tradeoff between increased model complexity and the extra error introduced by propagating through the additional requisite measurement and analysis steps. Given our data obtained by measuring the kinematics of the arm with motion tracking, it seems that the coordinate frame with which to best explain area 2 neural activity is simply the one with the most information about the arm kinematics and the fewest steps in processing. However, this does not rule out the idea that area 2 more nearly

represents a different whole-arm model that may be less abstracted from physiology, like musculotendon length or muscle spindle activity.

Still, this model comparison shows that even after proprioceptive signals reach area 2, neural activity can still be predicted well by a convergence of muscle-like signals, even though the signals have been processed by several sensory areas along the way. One potential explanation for this is that at each stage of processing, neurons simply spatially integrate information from many neurons of the previous stage, progressively creating more complex response properties. This idea of hierarchical processing was first used to explain how features like edge detection and orientation tuning might develop within the visual system from spatial integration of the simpler photoreceptor responses (*Felleman and Van Essen, 1991*; *Hubel and Wiesel, 1959*; *Hubel and Wiesel, 1962*). This inspired the design of deep convolutional artificial neural networks, now the state of the art in machine learning for image classification (*Krizhevsky et al., 2012*). Unlike previous image recognition methods, these feedforward neural networks are not designed to extract specific, human-defined features of images. Instead, intermediate layers learn to integrate spatially patterned information from earlier layers to build a library of feature detectors. In the proprioceptive system, such integration, without explicit transformation to some intermediate movement representation, might allow neurons in area 2 to serve as a general-purpose library of limb-state features, whose activity is read out in different ways for either perception or use in motor control.

## Hand kinematic-force model

Overall, our main results showed that the whole-arm model better captures firing rates and features of the neural activity than does the hand-only model. One consideration in interpreting these results is the fact that the whole-arm model is almost twice as expressive as the hand-only model, due to its greater number of parameters. While we took care to make sure the models were not overfitting (see Methods for details on cross-validation), a concern remains that any signal related to the behavior may improve the fits, simply because it provides more information. To address this concern, we would ideally compare these results with those from a model with the same number of parameters, but with behavioral signals uncorrelated with elbow kinematics, for example, kinematics of the other hand. Unfortunately, due to experimental constraints, we only collected tracking information from the reaching arm. As a substitute, we also tested a model we titled 'hand kinematic-force', which builds on the hand-only kinematic model by adding the forces and torques on the manipulandum handle. This model is similar to one proposed by *Prud'homme and Kalaska (1994)* and has the same number of parameters as the whole-arm model. While the handle forces and torques are likely correlated with the elbow kinematics, this model serves as a reasonable control to explore the particular importance of whole-arm kinematics to area 2.

*Appendix 1—figure 3* shows comparisons between the whole-arm model and the hand kinematic-force model on the three metrics we used. We found that the $pR^2$ and the tuning curve correlation values for both models were comparable, with some neurons better described by the whole-arm model and others by the kinematic-force model. However, we also found that the hand kinematic-force model often could not predict large changes in PD as well as the whole-arm model could (*Figure 7*; *Appendix 1—figure 3*). In four out of eight sessions, the whole-arm model had a significantly higher cVAF than the hand kinematic-force model. In the other sessions, there was no significant difference. While the two models made similar activity predictions, the better PD shift predictions suggest that the whole-arm model is a better model for area 2 neural activity.

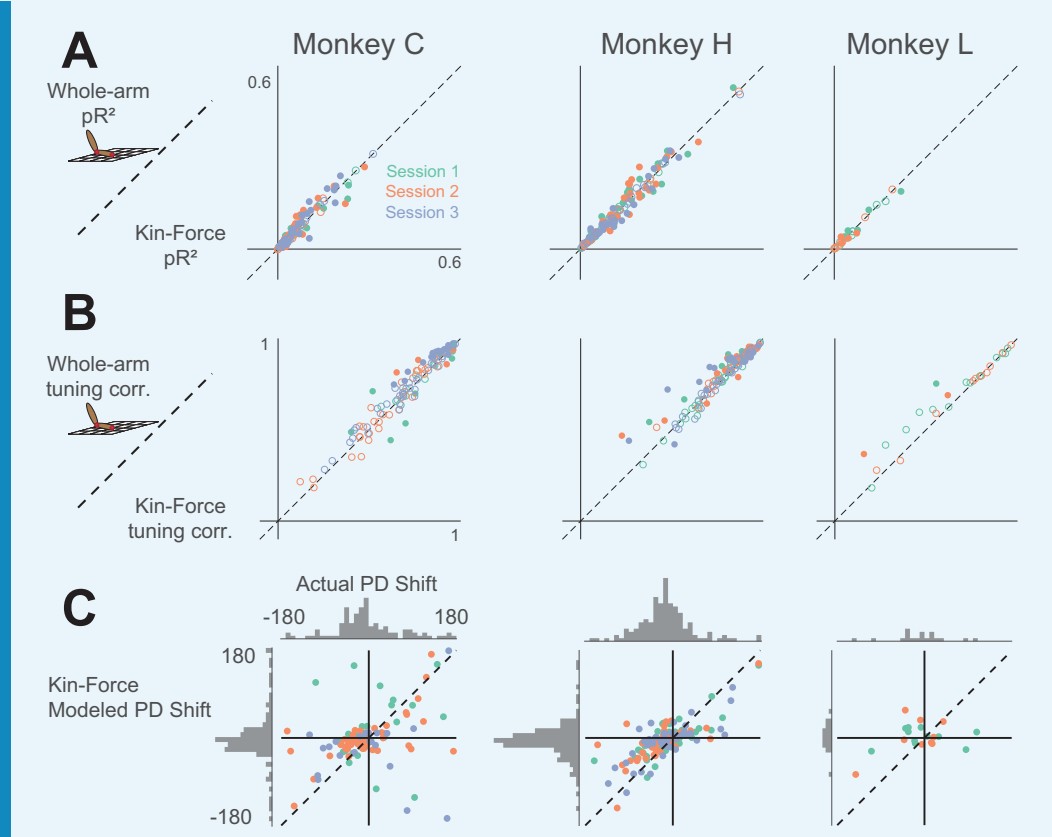

**Appendix 1—figure 3.** Comparison between whole-arm model and hand kinematic-force model (shortened as 'Kin-Force'). Same format as *Appendix 1—figures 1* and *2*.

