## [Decision Letter]

**Acceptance summary:**

The authors study the explanatory power of hand- and arm- kinematic model for proprioceptive processing in Brodmann Area 2. The authors designed two elegant paradigms to systematically vary hand and arm kinematics and examined neural responses to movement. They found that a whole arm model successfully predicted reaching-related limb state, but this model failed to characterise neuronal differences between active and passive movements. Cortical representation of proprioception is still poorly understood, and as such, the detailed investigation offered by the authors spanning multiple tasks and converging analyses is highly timely. The reviewers were particularly appreciative of the extensive revisions carried out by the authors, providing a more nuanced understanding of the power and limitations of the whole-arm model. Together, the reported results improve our understanding of how proprioceptive signals are represented in somatosensory cortex and open up new questions on the nature of active versus passive proprioceptive processing, with important implications for future development of proprioceptive feedback to brain machine interfaces.

**Decision letter after peer review:**

Thank you for submitting your article "Area 2 of primary somatosensory cortex encodes kinematics of the whole arm" for consideration by *eLife*. Your article has been reviewed by three peer reviewers, including Tamar R Makin as the Reviewing Editor and Reviewer #1, and the evaluation has been overseen by Joshua Gold as the Senior Editor.

The reviewers have discussed the reviews with one another and the Reviewing Editor has drafted this decision to help you prepare a revised submission.

In particular, the reviewers agreed that more detail relating to the somatosensory RF properties of the recorded neurons need to be considered. In addition, it is essential that more evidence needs to be presented to clarify the unique advantages of the more complex model (accounting for whole-arm position). Relatedly, the reviewers agreed that the authors need to consider alternative differences between the active and passive conditions that could have been picked up by the more complex model vicariously. Finally, it was felt that the outlook offered by the authors regarding the "classical" model of area 2 representation was not necessarily well accepted, requiring the authors to reframe the innovation offered in the present study.

Summary:

Here the authors demonstrate, over different paradigms and analyses, that activity in Brodmann area 2 can be best described when considering a combination of inputs for both hand and whole-arm position, rather than directional tuning, or muscle state. Cortical representation of proprioception is still poorly understood, and area 2 is considered a cortical proprioceptive gateway. As such, the detailed investigation offered by the authors spanning multiple tasks and converging analyses is highly timely in terms of methodology, basic understanding of proprioception and BMI applications. Moreover, by demonstrating that a given population in SI factors information of both hand and arm parameters, it challenges the canonical 'homunculus' account of SI, by which specific body parts are represented in separation to each other. While there was a lot to like in the paper, there were also some important confounds, some of which already brought up by the authors, that the reviewers felt needed to be addressed more comprehensively.

Essential revisions:

1) The reviewers raised multiple comments relating to the RF properties and the recording site. Reviewer 1 was questioning whether the extrinsic hand model is a scarecrow model – after all, the authors are recording activity from the S1 arm area. So it is not a great wonder that this area is more tuned to arm inputs than to the hand? In other words, perhaps if the authors recorded from the hand SI area, they might find greater invariance to arm position? They also asked for a confirmation that the authors are recording from area 2 and not area 5 (which is already considered to have a more complex higher-order representation)? This is especially important for the population analysis that could be driven by a subset of neurons with the most relevant information. Reviewer 2 asked for some evidence for electrodes location/RF properties relative to existing topographical map (e.g. Figure 19 of Pons et al., 1985). Both Reviewers asked that the authors demonstrated the implant location more accurately (the cartoon in Figure 1 leaves a lot to be desired). From the Materials and methods it appears that the authors did some classic assessment of RF properties, and all three reviewers wanted to learn more about the tuning properties, especially with respect to the hand and arm. Reviewer 3 asked whether the population of neurons that are better described by either egocentric/extrinsic or muscle or hand/elbow (Figure 6/7) are also separable by their hand mapped RF locations. Similarly, Reviewer 2 pointed that considering the broad receptive fields of area 2 (e.g. Pons et al., 1985) from leg to face representation (both deep and cutaneous), it is important to know how the kinematic representation of the arm (wrist and elbow) were related to the simple (passive) receptive field of each neuron. For example, does the neuron representing elbow kinematics possess the passive receptive fields of the hand? This would suggest the convergence of high-level representation onto area 2.

2) The reviewers felt that the explanation offered by the authors as for why the more complex model did a better job fitting the data was unsatisfying. The main finding is the contribution of elbow's kinematics for generating the neuronal firing patterns observed during movement. But as the authors discuss, the performance of the model should be improved by adding multiple (and different) potential covarying inputs. The issue at hand is not necessarily over-fitting (the cross-validation analysis is a reasonable way to demonstrate this), but rather the identification/characterisation of the variable that improved the model fit. The reviewers suggested a range of other factors that could have contributed to the differences between active/passive or egocentric/allocentric: Reviewer 1 suggested that the improved classification under the hand/elbow model be partially explained by the simple fact that here you are modelling a difference between active and passive? in other words, if you included a free parameter that would account for a main effect of active/passive, would the hand/elbow model still be able to outperform it? Reviewer 2 suggested that we could expect comparable improvement if other parameters were fitted, like grip force, shoulder or trunk position, or even the kinematics of another arm. Reviewer 3 asked whether, instead of hand – egocentric/extrinsic elbow reference frame, a wrist or shoulder reference frame has predictive similar value? Also, during the second behavioural task, were kinematic reference frames better at accounting for the neuronal firing during specific phases of the movement? As such, the current model comparison leaves a lot to be desired, and the reviewers felt that the main argument offered by the authors was not sufficiently conclusive. Reviewer 2 kindly offered potential practical solutions to improve the present evidence. For example, the authors could add another model evaluating solely the contribution of elbow and shoulder position. This is comparable with the model of "extrinsic" and "egocentric" in light of the model's complexity. By finding higher performance with the elbow model, for example, the authors could suggest a considerable contribution of proximal joints. An additional approach would be to examine the difference between hand and hand/elbow model further. It looks that the performance of two models is comparable (see 6-8 sec in Figure 5C) during some movement but not the other (3-4 sec in the same figure). How was the kinematics of hand and elbow during those periods? How did they covary? Adding a 'proper' control – shuffled noise and kinematics that covaried with endpoint but less important than elbow (e.g. shoulder, another limb) may solve this problem.

3) Relatedly, the reviewers agreed that alternative interpretations should be explored/ruled out for the observed differences between the active and passive conditions. Reviewer 1 pointed that from the example in Figure 3A it seems that activity (no. of spikes) was stronger for active vs. passive trials. If so, could this account for some of the key differences, e.g. the linear boundary between active and passive movements found in the PCA? If so, could this be explained by trivial task differences (e.g. differences in attention could modulate overall activity levels)? More interestingly, could this 'main effect' of activity (that I am postulating here) be driven by motor inputs? If so, it might have a unique time-stamp? This was also echoed in Reviewer 2's point that the major difference between active and passive movement is the existence of "corollary discharge" or "efference copy", so the complex model may actually decode this in addition to the endpoint. Reviewer 2 also pointed that this result may suggest that the extent of sensory cancellation by ego-motion is different between the control of endpoint or more proximal joints. If so, can the authors link this implication with the proposal of whole arm representation in area 2?

4) Finally, while the reviewers found the manuscript to offer important contribution and innovation by exploring the role of whole-arm position in proprioceptive representation, they felt the conceptual framework where this contribution is pitted against a 'classical' model was overly simplistic and unappealing. First, the portrayal of area 2 as an area where 'neural activity is related simply to the movement of the hand and interaction forces it encounters during movement, similar to our conscious experience of proprioception' seems over-stated. The authors cite several previous studies who suggested movement direction encoding in S1 in monkeys (most prominently by the Miller group), but the reviewers felt that even when considering these citations, this statement is far from the consensus for how proprioception is represented in area 2. Even though, historically, endpoint kinematics was thought to be the guiding principle in motor cortex, this preconception is not widely held in area 2. As such, the reviewers agreed that the null hypothesis of this paper (Abstract) is not strong enough, and requires rethinking. The manuscript should therefore be substantially re-written to highlight the novel contributions of the empirical work in characterising the neuronal properties in this area, independently of the end-point model.

---

## [Author Response]

Essential revisions:1) The reviewers raised multiple comments relating to the RF properties and the recording site. Reviewer 1 was questioning whether the extrinsic hand model is a scarecrow model – after all, the authors are recording activity from the S1 arm area. So it is not a great wonder that this area is more tuned to arm inputs than to the hand? In other words, perhaps if the authors recorded from the hand SI area, they might find greater invariance to arm position?

First, we would like to clarify that the hand kinematics we refer to are the kinematics of translation of the hand through space, not motion of the fingers. Whereas we would expect to find the latter in the hand area of area 2, we likely would not find the former.

This hand-only model has typically been used when studying the arm representation of area 2. As with the classic reaching studies of M1 beginning with Georgopoulos, its appeal is its reasonable accuracy despite its simplicity. Our main purpose is to determine how much additional information we might learn about area 2 activity by considering the movement of the whole arm. We have restructured the paper to reflect this purpose. To achieve this, we first swapped the order in which we describe the experiments, showing first that we gain explanatory power during the two-workspace task by considering the movements of the whole arm. Second, we reanalyzed the data from the active/passive experiment and rewrote the corresponding section so as not to assume the classic hand-only model as a null hypothesis. We describe the changes to the active/passive experiment analyses in our response to point 3.

They also asked for a confirmation that the authors are recording from area 2 and not area 5 (which is already considered to have a more complex higher-order representation)? This is especially important for the population analysis that could be driven by a subset of neurons with the most relevant information. Reviewer 2 asked for some evidence for electrodes location/RF properties relative to existing topographical map (e.g. Figure 19 of Pons et al., 1985). Both Reviewers asked that the authors demonstrated the implant location more accurately (the cartoon in Figure 1 leaves a lot to be desired). From the Materials and methods it appears that the authors did some classic assessment of RF properties, and all three reviewers wanted to learn more about the tuning properties, especially with respect to the hand and arm. Reviewer 3 asked whether the population of neurons that are better described by either egocentric/extrinsic or muscle or hand/elbow (Figure 6/7) are also separable by their hand mapped RF locations. Similarly, Reviewer 2 pointed that considering the broad receptive fields of area 2 (e.g. Pons et al., 1985) from leg to face representation (both deep and cutaneous), it is important to know how the kinematic representation of the arm (wrist and elbow) were related to the simple (passive) receptive field of each neuron. For example, does the neuron representing elbow kinematics possess the passive receptive fields of the hand? This would suggest the convergence of high-level representation onto area 2.

We have now added to the manuscript more detail on the implant location, along with results from sensory mappings on the arrays (now Figure 1). In our mappings, we found a rough gradient across the arrays of Monkeys C and H corresponding to a somatotopy from proximal to distal receptive fields on the arm, which agrees with Figure 19 of Pons et al., 1985. Further, we found a mixture of cutaneous and deep receptive fields across our arrays, which matches descriptions of area 2 from previous anatomical studies (Seelke et al., 2011).

Unfortunately, we did not perform sensory mappings around the same time periods as we collected the behavioral data. As such, we cannot confidently match up neural receptive field properties with model performance. However, there was no link between model performance in the two-workspace experiment and electrode location on the array, suggesting that the whole-arm model predictions of area 2 activity did not depend on the exact receptive field of the recorded neurons. It is not clear to us what it would mean for a neuron with elbow responses to have a hand receptive field. In any case, neurons did not typically have this kind of extended, combined receptive field.

2) The reviewers felt that the explanation offered by the authors as for why the more complex model did a better job fitting the data was unsatisfying. The main finding is the contribution of elbow's kinematics for generating the neuronal firing patterns observed during movement. But as the authors discuss, the performance of the model should be improved by adding multiple (and different) potential covarying inputs. The issue at hand is not necessarily over-fitting (the cross-validation analysis is a reasonable way to demonstrate this), but rather the identification/characterisation of the variable that improved the model fit. The reviewers suggested a range of other factors that could have contributed to the differences between active/passive or egocentric/alocentric: Reviewer 1 suggested that the improved classification under the hand/elbow model be partially explained by the simple fact that here you are modelling a difference between active and passive? in other words, if you included a free parameter that would account for a main effect of active/passive, would the hand/elbow model still be able to outperform it? Reviewer 2 suggested that we could expect comparable improvement if other parameters were fitted, like grip force, shoulder or trunk position, or even the kinematics of another arm. Reviewer 3 asked whether, instead of hand – egocentric/extrinsic elbow reference frame, a wrist or shoulder reference frame has predictive similar value? Also, during the second behavioural task, were kinematic reference frames better at accounting for the neuronal firing during specific phases of the movement? As such, the current model comparison leaves a lot to be desired, and the reviewers felt that the main argument offered by the authors was not sufficiently conclusive. Reviewer 2 kindly offered potential practical solutions to improve the present evidence. For example, the authors could add another model evaluating solely the contribution of elbow and shoulder position. This is comparable with the model of "extrinsic" and "egocentric" in light of the model's complexity. By finding higher performance with the elbow model, for example, the authors could suggest a considerable contribution of proximal joints. An additional approach would be to examine the difference between hand and hand/elbow model further. It looks that the performance of two models is comparable (see 6-8 sec in Figure 5C) during some movement but not the other (3-4 sec in the same figure). How was the kinematics of hand and elbow during those periods? How did they covary? Adding a 'proper' control – shuffled noise and kinematics that covaried with endpoint but less important than elbow (e.g. shoulder, another limb) may solve this problem.

We thank the reviewers for their many different suggestions for model comparisons that could make our arguments stronger. We have chosen two additional models to add to the supplementary information to strengthen our case that the whole-arm kinematics do matter, rather than just being epiphenominal. These model comparisons help to demonstrate the particular covariates that matter most for predicting neural activity in area 2. These include an extension of the hand-only model with added forces and torques acting on the manipulandum handle. This model has 13 parameters, which gives it the same level of complexity as the hand/elbow model (now called the whole-arm model for simplicity). While the two models had similar performance in pR^2^ and tuning curve correlation, the force-augmented model could not predict PD shifts as well as the whole-arm model. We have also added a joint kinematic model, which relates neural activity to the angles and angular velocities of the 7 degrees of freedom in the arm, with even more parameters than the whole-arm model. Like the muscle model, this joint model performed worse than the whole-arm model.

Unfortunately, we are not sure what type of “shuffled” control might be appropriate. While the amount of information added by a movement-related signal could be reduced by shuffling it, it is not clear to us to what end that would be done.

We agree that it would be interesting to look at how hand and elbow kinematics covary at points when the models make similar or different predictions. Unfortunately, because our task involved reaching to random targets, where trials were never repeated, such a covariance analysis would be difficult to perform and subsequently interpret. However, one might imagine a follow-up experiment in which we identify sequences of targets that result in either similar or different predictions and have the monkey repeat these sequences at random, in order to adequately analyze the covariance of hand and elbow in these different cases.

As noted above, and detailed below, we have heavily revised the active/passive section of our paper. While we did not explicitly add a free parameter to the model, we rather carefully explored the whole-arm model’s dependence on this characteristic of movement. The new section demonstrates that whole-arm kinematics are not always sufficient to explain area 2 activity, which also acts as a control for the analyses of the two-workspace section.

3) Relatedly, the reviewers agreed that alternative interpretations should be explored/ruled out for the observed differences between the active and passive conditions. Reviewer 1 pointed that from the example in Figure 3A it seems that activity (no. of spikes) was stronger for active vs. passive trials. If so, could this account for some of the key differences, e.g. the linear boundary between active and passive movements found in the PCA? If so, could this be explained by trivial task differences (e.g. differences in attention could modulate overall activity levels)? More interestingly, could this 'main effect' of activity (that I am postulating here) be driven by motor inputs? If so, it might have a unique time-stamp? This was also echoed in Reviewer 2's point that the major difference between active and passive movement is the existence of "corollary discharge" or "efference copy", so the complex model may actually decode this in addition to the endpoint. Reviewer 2 also pointed that this result may suggest that the extent of sensory cancellation by ego-motion is different between the control of endpoint or more proximal joints. If so, can the authors link this implication with the proposal of whole arm representation in area 2?

Based on these insightful comments, we revisited the data from our active/passive experiment, exploring more thoroughly how well a whole-arm model explains how area 2 represents these two types of movement. In doing so, we found that while the whole-arm model could explain separability well, it was not actually capturing a consistent relationship between neural activity and arm kinematics for many of the neurons. This finding suggests that the separation in area 2 activity between active and passive movements cannot be explained purely by arm kinematics. Efference copy must certainly be considered as an explanation, as the reviewers pointed out. We have rewritten this section and added a section to the Discussion to reflect these findings.

4) Finally, while the reviewers found the manuscript to offer important contribution and innovation by exploring the role of whole-arm position in proprioceptive representation, they felt the conceptual framework where this contribution is pitted against a 'classical' model was overly simplistic and unappealing. First, the portrayal of area 2 as an area where 'neural activity is related simply to the movement of the hand and interaction forces it encounters during movement, similar to our conscious experience of proprioception' seems over-stated. The authors cite several previous studies who suggested movement direction encoding in S1 in monkeys (most prominently by the Miller group), but the reviewers felt that even when considering these citations, this statement is far from the consensus for how proprioception is represented in area 2. Even though, historically, endpoint kinematics was thought to be the guiding principle in motor cortex, this preconception is not widely held in area 2. As such, the reviewers agreed that the null hypothesis of this paper (Abstract) is not strong enough, and requires rethinking. The manuscript should therefore be substantially re-written to highlight the novel contributions of the empirical work in characterising the neuronal properties in this area, independently of the end-point model.

We are glad that the reviewers appreciate the work presented in this paper. As mentioned above, we have restructured the manuscript to highlight our novel contributions in characterizing area 2. Now, instead of framing the paper as a study of how the hand-only model fails, we frame our results in the more positive light of showing what we can learn about S1 representations by tracking the whole arm.